



# Suitability of fiber-optic distributed temperature sensing to reveal mixing processes and higher-order moments at the forest-air interface

Olli Peltola[1], Karl Lapo[2,3], Ilkka Martinkauppi[4], Ewan O'Connor[1,5], Christoph K. Thomas[2,3], and Timo Vesala[6,7]

[1]Climate Research Programme, Finnish Meteorological Institute, P.O. Box 503, 00101 Helsinki, Finland
[2]Micrometeorology Group, University of Bayreuth, Bayreuth, Germany
[3]Bayreuth Center for Ecology and Environmental Research, Bayceer, University of Bayreuth, Bayreuth, Germany
[4]Geological survey of Finland, Kokkola, Finland
[5]Department of Meteorology, University of Reading, Reading, UK
[6]Institute for Atmosphere and Earth System Research/Physics, Faculty of Science, University of Helsinki, P.O. Box 68, 00014 Helsinki, Finland
[7]Institute for Atmospheric and Earth System Research/Forest Sciences, Faculty of Agriculture and Forestry, University of Helsinki, P.O. Box 27, 00014, Helsinki, Finland

**Correspondence:** Olli Peltola (olli.peltola@fmi.fi)

**Abstract.** Suitability of fibre-optic distributed temperature sensing (DTS) technique to observe atmospheric mixing profiles within and above forest was quantified, and these profiles were analysed. The spatially continuous observations were made at a 125 m tall mast in a boreal pine forest. Air flows near forest canopies diverge from typical boundary layer flows due to the influence of roughness elements (i.e. trees) on the flow. Ideally these complex flows should be studied with spatially continuous

measurements, yet such measurements are not feasible with conventional micrometeorological measurements with e.g. sonic anemometers. Hence the suitability of DTS measurements for studying canopy flows was quantified.

The DTS measurements were able to discern continuous profiles of turbulent fluctuations and mean values of air temperature along the mast providing information about mixing processes (e.g. canopy eddies, evolution of inversion layers at night) and up to third order turbulence statistics across the forest-atmosphere interface. Turbulence measurements with 3D sonic anemome-

10 ters and Doppler lidar at the site were also utilised in this analysis. The continuous profiles for turbulence statistics were in line with prior studies made at wind tunnels and large eddy simulations for canopy flows. The DTS measurements contained a significant noise component which was however quantified and its effect on turbulence statistics was accounted for. Underestimation of air temperature fluctuations at high frequencies caused 20...30 % underestimation of temperature variance at typical flow conditions. Despite these limitations, the DTS measurements should prove useful also in other studies concentrating on

flows near roughness elements and/or non-stationary periods, since the measurements revealed spatio-temporal patterns of the flow which were not possible to discern from single point measurements fixed in space.



## 1 Introduction

The majority of the interaction between the atmosphere and Earth's surface takes place in a shallow air layer termed the atmospheric boundary layer (ABL). Insights on the atmospheric mixing processes in this layer are required in order to gain a better understanding on ecosystem-atmosphere feedbacks, air quality and weather forecasting related issues. Studies near the
surface typically rely on time series analysis, since spatial details of the mixing close to the ground are difficult to measure from a practical point-of-view. Hence, spatial statistics are typically derived from a time series of observations by assuming Taylor's frozen turbulence hypothesis Taylor (1938), yet it is recognised that this assumption is not universally valid (Mahrt et al., 2009; Thomas, 2011; Higgins et al., 2012; Cheng et al., 2017). This discrepancy calls for spatially explicit measurements. Another motivation arises from the fact that similarity theories underlying the analysis of observations and models are posed in length
scales.

Atmospheric boundary layer flows feature other flow modes besides turbulence and these flow modes exhibit spatial patterns which are not directly related to their temporal counterparts. For instance, transient submeso-scale motions may occur in the weak-wind stable boundary layer (Mahrt, 2014), which can travel in the opposite direction of the mean flow (Zeeman et al., 2015) and interact with turbulence (Kang et al., 2015; Sun et al., 2015; Mahrt and Thomas, 2016; Vercauteren et al., 2016)
inflicting intermittent mixing and transport of gases, heat and momentum. Mechanistic understanding of these non-turbulent motions and related processes are limited, partly due to the missing spatial information of the flow.

Besides submeso motions in stable conditions, the flow in the roughness sublayer (RSL) by definition exhibits spatial patterns that call for spatial sampling. For instance, flows within and above forest canopies are dominated by large coherent structures (Finnigan, 2000; Thomas and Foken, 2007a, b; Thomas et al., 2008; FINNIGAN et al., 2009) that are in continuous interaction
with the surface roughness elements; this interaction can cause persistent spatial variability in the flow and turbulent transport (Bohrer et al., 2009; Schlegel et al., 2015) raising questions about the experimental approach deployed in global measurement networks monitoring the turbulent transport of gases above ecosystems (Baldocchi, 2014). Surface spatial heterogeneities dominate the flow properties also in urban RSL, with urban surfaces exhibiting large variability in the surface heat flux and hence the thermal production of turbulence (Barlow, 2014). In general, the analysis of the effect of abrupt edges, or irregular
discontinuities in surface characteristics, on the flow requires spatially explicit sampling.

Remote sensing instrumentation is capable of resolving the spatial detail of the flow (e.g. Newsom et al., 2008; TRÄUM­NER et al., 2012), yet small-scale features close to the ground are typically missing and measurements near obstacles such as trees and buildings are not feasible. On the other hand, conventional precise in-situ measurements circumvent these limitations yet are fixed in space (e.g. at measurement towers) and hence spatial details of the flow can be deduced only by assuming
Taylor's frozen turbulence hypothesis. Hence there is an evident observational gap between the conventional measurement techniques (remote sensing vs. in-situ) which results in limited understanding of flow processes falling in this observational gap. In-situ measurements on moving platforms, such as tethered balloons or unmanned aerial vehicles (UAVs) (Poulos et al., 2002; Frehlich et al., 2008; Higgins et al., 2018; Egerer et al., 2019), may partly fill the gap, yet with these techniques only non-continuous campaign type measurements are possible resulting in low temporal coverage and representativeness. Further-



more, measurements close to or in-between roughness elements (trees, buildings) are typically not possible with these mobile platforms.

Distributed temperature sensing (DTS) has been utilised in environmental research since the first studies of Selker et al. (2006) and Tyler et al. (2009). The measurement method provides spatially continuous profiles along a fibre-optic cable which

can be freely distributed in the measurement domain. The DTS data are provided at similar temporal and spatial resolution along the cable as conventional in-situ measurements lending direct comparison between the measurement techniques and complementary analyses of joint measurements with multiple techniques. The measurement technique relies on optical time-domain reflectometry and measurement of Raman back-scattering of a light pulse traveling in the fibre-optic cable (Dakin et al., 1985; Selker et al., 2006). Due to its ability to provide spatio-temporal information at scales (down to 1 s and 25 cm)

that are commensurate with the scales prevalent near the surface, the measurement method shows promise in answering many persistent unanswered questions related to near surface flow.

A growing body of research has already utilised DTS measurements in atmospheric studies (Keller et al., 2011; Thomas et al., 2012; Euser et al., 2014; de Jong et al., 2015; Sayde et al., 2015; Zeeman et al., 2015; Pfister et al., 2017; Higgins et al., 2018; Schilperoort et al., 2018; Higgins et al., 2019; Izett et al., 2019; Mahrt et al., 2019; Pfister et al., 2019). Bulk of the

studies have concentrated on nocturnal flows near the surface (Keller et al., 2011; Thomas et al., 2012; Zeeman et al., 2015; Pfister et al., 2017; Izett et al., 2019; Mahrt et al., 2019; Pfister et al., 2019) and a few have concentrated on transition periods (Higgins et al., 2018, 2019). By utilising different DTS measurement configurations, some studies have done distributed wind speed (heated cables) (Sayde et al., 2015; Pfister et al., 2017) or humidity (wetted cables) measurements (Euser et al., 2014; Schilperoort et al., 2018), whereas others have examined the radiation error of the cables (de Jong et al., 2015; Sigmund et al.,

2017). However, thorough and critical analysis on the feasibility of DTS system to measure atmospheric scalar mixing has not been done since the pioneering study of Thomas et al. (2012). We complement the analysis made by Thomas et al. (2012) by comparing the DTS measurements against conventional in-situ analysers within and above an aerodynamically rough forest canopy from the ground up to 120 m aloft. Furthermore, we evaluate whether also third-order statistics could be discerned from the DTS data revealing important transport process information of the sweep-ejection cycles created by the coherent structures.

Deviations from Gaussian distribution are typically related to large coherent eddies or submeso air motions, whereas isotropic turbulence follows more closely Gaussian distribution and hence higher order statistics are needed for studying these more organised flow patterns. We also assess the random uncertainties in the DTS measurements and their effect on the derived statistics. Finally we demonstrate how a combination of vertical DTS measurements together with 3D sonic anemometers and upward pointing Doppler lidar measurements can be used to obtain continuous turbulence profiles starting from canopy

sublayer all the way up to boundary layer top and how DTS system bridges the scales between individual in-situ measurement devices and remote sensing with lidar.



## 2 Materials and methods

### 2.1 Measurement site and instrumentation

The measurement campaign took place from 3 June to 8 July 2019 at the Hyytiälä SMEAR II station (Hari and Kulmala, 2005) located in central Finland (61.845°N, 24.289°E). The station is a member of several measurement networks, including

ICOS (Integrated Carbon Observing System; Franz et al. (2018)) and ACTRIS (Aerosols, Clouds, and Trace gases Research Infrastructure), and thus a wide range of atmospheric and biospheric measurements are conducted continuously. The site is dominated by a Scots pine (*Pinus Sylvestris*) stand with average tree height ($h$) of approximately 17 m resulting in zero-plane displacement height ($d$) of 14 m. The forest canopy is between 10 and 17 m and the all-sided leaf area index (LAI) was approximately 8 $m^2$ $m^{-2}$ in 2015. Below 10 m height there is relatively open trunk space. The terrain is undulating with the

main slope (2°) directed roughly in the north-south direction. More details on the in-canopy turbulence and canopy structure are given in Launiainen et al. (2007) and on the terrain undulation in Alekseychik et al. (2013).

The measurements utilized in this study were conducted on the SMEAR II station 125 m tall main mast. 3D ultrasonic anemometers are installed at five heights on the mast (5.5, 24.6, 27, 68.3 and 125 m above ground) providing turbulent fluctuations of air temperature and three wind speed components at 10 Hz sampling frequency. Different anemometers are present

at different heights (USA-1 manufactured by METEK GmbH, Germany, at 5.5, 68.3 and 125 m; Solent Research 1012R2 by Gill Ltd., UK, at 24.6 m; HS-50 by Gill Ltd., UK, at 27 m). The anemometer at 27 m height is part of the eddy covariance (EC) flux measurement setup for ICOS measurements (Rebmann et al., 2018). Measurements at 5.5 m height started on 18 June 2019, all other 3D sonic anemometers were measuring continuously throughout the study period. The air temperature vertical profile below the ICOS EC measurement level was measured following ICOS procedures (Montagnani et al., 2018).

Radiation shielded and ventilated platinum-wire thermistors (PT-100) were located at 3.3, 5.8, 8.8, 12.5, 16.8, 21.6 and 27 m heights above the ground. These measurements provide suitable data for comparison against the temperature data measured with DTS. Radiation components were measured with a four-component net radiometer (Model CNR4, Kipp & Zonen, Delft, Netherlands) mounted on the mast at 67 m height.

Besides conventional in-situ measurements on the mast, the atmospheric boundary layer was profiled with a scanning

Doppler wind lidar (Halo Photonics Streamline) (Pearson et al., 2009) located on the roof of a building approximately 400 m South-West from the mast. The wind lidar operates at 1.5 $\mu$m wavelength and, when pointing vertically, is configured to provide a profile of radial velocities approximately every 14 seconds with 30 m range resolution at Hyytiälä (Hirsikko et al., 2014).

### 2.1.1 DTS measurement setup

The DTS instrument (Ultima-S, 5 km variant, Silixa Ltd., Hertfordshire, UK) was housed in a wooden cabin located approximately 30 m away from the measurement mast. Two 50 l calibration baths were filled with water, equipped with reference thermometers (PT-100; supplied with the DTS instrument and logged with the DTS instrument at 0.01 °C precision) and the fibre-optic cable was guided twice through both baths. The baths were also equipped with aquarium pumps to ensure continu-





ous mixing and prevent stratification of the water. Water temperature in the baths was controlled with two thermostats (RC 6 CS, LAUDA DR. R. WOBSER GMBH & CO. KG, Lauda-Königshofen, Germany) and the bath temperatures were set to 5 °C and 30 °C so that the temperatures bracketed the ambient air temperatures during the campaign. The baths were located next to the wooden cabin which housed the DTS instrument.

The measurements were conducted in double-ended mode with signals from both directions stored separately (see Sect. 2.2 for processing). The path of the fibre-optic cable began by running through both calibration baths, then up and down the 125 m tall mast, and returning through the calibration baths again to finish. The cable was fastened to approximately 0.5 m long horizontal booms located every 5 to 10 meters along the mast by squeezing the cable gently between two metallic plates covered with rubber slabs and soft electrical tape. Despite careful mounting of the cable, the cable holders caused disturbances

in the data and consequently data points close to the holders were removed from subsequent analysis. Horizontal separation between the cable and the 3D sonic anemometers was 3.5 m at all levels. The vertical DTS measurements extended from 2 m to 120 meters above the ground. This setup enabled the provision of reference measurement locations at both the beginning and end of the cable from the calibration baths, and provide a double measurement of the DTS vertical air temperature profile at 12.7 cm spatial and 1 Hz temporal resolution. Note, however, that in double-ended mode, each direction along the cable is

measured sequentially, so that the temperature data was available at 0.5 Hz resolution. A white, thin (outer diameter 0.9 mm) aramid reinforced 50 $\mu$m multimode fibre-optic cable (AFL Telecommunications, Duncan, SC 29334, US) was utilised in this study in order to minimise the impact of solar heating on the measurements (de Jong et al., 2015; Sigmund et al., 2017) and to maximise the high frequency response of the setup.

When comparing the results from this study to Thomas et al. (2012) it is important to note that they used a different variant

of the Ultima-S instrument, which was an older model optimised for shorter cable lengths (2 km) utilising different laser and detector components. The noise floor of this older variant used in Thomas et al. (2012) is smaller by a factor of 2 to 3. Hence, differences in instrument performance are to be expected, foreshadowing the results.

## 2.2 Data processing

All high frequency time series (3D sonic anemometers and DTS) were separated into the slowly varying mean and fluctuations

around the mean:

$$T = T' + \overline{T}, \tag{1}$$

where T is the measured high frequency time series, overbar denotes averaging in time, and prime (') fluctuations around the mean calculated as a residual ($T' = T - \overline{T}$). Processing of 3D sonic anemometer data followed commonly applied procedures. The data were despiked in order to remove spurious outliers from the time series (Mauder et al., 2013) and the wind vectors were

aligned with the long-term mean wind field following the planar-fit coordinate rotation method (Wilczak et al., 2001). Turbulent fluxes and statistics (variance, skewness) were calculated with 30-min resolution throughout this study, unless otherwise noted. The stability of the atmospheric surface layer was evaluated based on the stability parameter:

$$\zeta = \frac{z - d}{L}, \tag{2}$$





where $z$ denotes height, $d$ displacement height, and $L$ Obukhov length. Negative values for $\zeta$ denote unstable stratification (buoyancy increases turbulence), positive values denote stable stratification (buoyancy diminishes turbulence) and zero denotes neutral stratification. A scaling parameter for temperature was used to describe the temperature variability in the atmospheric surface layer, and was defined as

$$T_* = \frac{\overline{w'T'}}{u_*}, \tag{3}$$

where $u_*$ is friction velocity. This value was calculated using data from the EC system at 27 m height throughout the study.

DTS measurements were post-field calibrated using the measurements in the calibration baths and the following equation (e.g. van de Giesen et al. (2012)):

$$T(x,t) = \frac{\gamma}{\ln\left(\frac{P_s(x,t)}{P_{as}(x,t)}\right) + C(t) + \int_0^x \Delta\alpha(x')dx'}, \tag{4}$$

where $T$ is the calibrated temperature at a function of position along the fibre ($x$) and time ($t$), $\gamma$ is a DTS system specific constant, $P_s$ and $P_{as}$ are the measured Stokes and anti-Stokes signals, $C$ is a time-dependent calibration parameter, and the integral in the denominator is related to the differential attenuation of the anti-Stokes and Stokes signals along the cable. The double-ended configuration allowed the determination of the differential attenuation at each location along the cable separately, which was calculated following van de Giesen et al. (2012) separately for each 30-min averaging period. However, measurements from only one direction were utilised to calculate the temperature along the cable, i.e. measurements from the two directions were not averaged. After determining the differential attenuation, the calibration parameters $\gamma$ and $C$ were first determined by fitting Eq. 4 to the reference measurements made in the calibration baths for each time step separately producing time series for both; then, during the second processing stage, the data were calibrated again by fixing the value of $\gamma$ to 476 $K$ (mean of values obtained during the first processing stage) and letting $C$ vary between time steps. We opted to process the data with this three-step procedure since 1) it resulted in less noisy calibration parameters, 2) theoretically $\gamma$ should be constant and 3) a reduction of fit parameters is desirable given the number of calibration baths. The calibration parameter $C$ showed only a slight variation in time ($1.460 \pm 0.003$ (mean $\pm$ SD) which resulted from a parabolic dependence on indoor temperature of the cabin housing the DTS instrument. After quality filtering, 1353 30-min periods of DTS data were available for further analysis.

The high frequency response of the DTS system was evaluated by transforming the temperature time series to frequency domain with Fourier transform and comparing the power spectra against reference power spectra measured with the sonic anemometers using the EC technique. Following EC data processing procedures (e.g. Ibrom et al. (2007)) the noise in the power spectra was subtracted from the signal prior to the comparison. By assuming that the DTS system as a whole behaves as a first-order response sensor, then, after normalisation, the ratio between the power spectra from DTS and the reference can be approximated to follow

$$T_H(f) = \frac{1}{1 + (2\pi f \tau)^2}, \tag{5}$$

where $f$ is frequency and $\tau$ is a first-order response time describing the high-frequency response of the system. An estimate for the attenuation of the DTS-derived temperature variance due to imperfect high frequency response and lower sampling





frequency was estimated using

$$AF = \frac{\int_{f_1}^{f_2} T_H S_{TT,low} \, df}{\int_{f_1}^{f_3} S_{TT} \, df}, \tag{6}$$

where $AF$ is the attenuation factor, $S_{TT}$ is the power spectrum measured with the EC system, $S_{TT,low}$ is the power spectrum calculated from EC data that was averaged in the time domain to match the temporal resolution of DTS prior to Fourier transform, and $T_H$ was calculated using a value for $\tau$ obtained experimentally with Eq. (5) (see Sect. 3.2). The integration limits $f_1$, $f_2$ and $f_3$ are set by the length of the averaging period and the Nyquist frequency for DTS and 3D sonic anemometer data, respectively. By definition, $AF = 1$ for a perfect sensor ($\tau = 0\,s$) with high sampling frequency (i.e. $f_2 = f_3$) and $AF = 0$ for a fully damped signal ($\tau \to \infty\,s$).

### 2.2.1 Determination of instrument noise

Instrument noise can significantly hinder the analysis of measurements if the signal to noise ratio is small. Here we utilised the methodology developed by Lenschow et al. (2000) to extract turbulent statistics from noisy data. The method originally developed for lidar data has been used also for EC data (Mauder et al., 2013; Rannik et al., 2016) and is part of ICOS EC data processing routines (Nemitz et al., 2018). The fluctuating part of the time series $T'(t)$ can be thought to consist of signal $T'_s(t)$ and noise $\epsilon(t)$ ($T' = T'_s + \epsilon$). All the time series $T'$, $T'_s$ and $\epsilon$ have zero means. The second-order autocovariance ($M_{11}$) can then be written as

$$M_{11}(t_l) = \overline{(T'_s + \epsilon)(T'_{sl} + \epsilon_l)}, \tag{7}$$

where overbar denotes time averaging and subscript $l$ denotes that the time series has been lagged with time lag $t_l$ (Lenschow et al., 2000). Assuming $\epsilon$ and $T'_s$ are uncorrelated with each other, it can be found that

$$M_{11}(0) = \overline{T'^2_s} + \overline{\epsilon^2}, \tag{8}$$

meaning that the time series variance $M_{11}(0)$ is a sum of signal variance $\overline{T'^2_s}$ and noise variance $\overline{\epsilon^2}$. Since noise is assumed to be uncorrelated, noise only contributes to the autocovariance at lag zero; hence the signal variance can be estimated by extrapolating the autocovariance values to zero lag and the noise variance can be estimated from the residual:

$$\overline{T'^2_s} = M_{11}(t_l \to 0) \tag{9}$$
$$\overline{\epsilon^2} = M_{11}(0) - M_{11}(t_l \to 0). \tag{10}$$

Similarly the effect of noise on third-order moments can be estimated using third-order autocovariance $M_{21}$:

$$\overline{T'^3_s} = M_{21}(t_l \to 0) \tag{11}$$
$$\overline{\epsilon^3} = M_{21}(0) - M_{21}(t_l \to 0), \tag{12}$$





where it was further assumed that the product of $T'_s$ to any power and $\epsilon$ to an odd power are zero (Lenschow et al., 2000). The time series skewness (Sk) was then estimated using the second- and third-order moments calculated with the equations above (Sk $= \frac{\overline{T'^3_s}}{(\overline{T'^2_s})^{3/2}}$). Signal-to-noise ratio was used to estimate the magnitude of noise relative to the signal:

$$\text{SNR} = \frac{\overline{T'^2_s}}{\overline{\epsilon^2}}. \tag{13}$$

## 5  3  Results and discussion

### 3.1  DTS instrument noise determination

DTS measurements at high spatial and temporal resolution contain a significant noise contribution due to the small number of backscattered Raman-photons, hence, duplicate or triplicate measurements are often conducted. In this study, the instrument noise was estimated using Lenschow et al. (2000) (see Sect. 2.2.1) for each location along the fibre for each 30-min averaging

period. Figure 1 shows the average dependence of the noise standard deviation, $\sigma_{noise}$, with length along the fibre, LAF, together with the variability observed in the calibration baths. The noise increases exponentially with LAF as expected since the number of photons available for backscattering decreases along the fibre. After taking the mean of the up-mast and down-mast measurements, and then averaging over four consecutive points, the average noise decreased from 0.62 K to 0.26 K, and was, in practise, independent of position along the fiber and thus height. This decrease is similar to what would be expected

for fully independent measurements (i.e. $0.62\text{K}/\sqrt{8} = 0.22\text{K}$). This averaging procedure was applied to all further analysis in this study and provides a temperature profile with approximately 0.5 m resolution from 2 m up to 120 m above the ground. No temporal averaging was applied in order to preserve the temporal resolution of the signal. The noise level after averaging was below the typical temperature variability measured above the forest canopy during the campaign.

Based on the fit shown in Fig. 1, the noise at zero LAF was approximately 0.57 K, close to the instrument specifications.

However, $\sigma_{noise}$ increased faster as a function of LAF than expected, likely due to experimental setup which may have caused mechanical strain onto the fibre (cable holders, wind load) and hence loss of signal. There were a few step losses (i.e. sudden decreases of $P_s$ and $P_{as}$, see also Eq. 4) along the cable caused by the support structures and these step losses added to the increases in the noise. The latter was linearly proportional to the magnitude of the step loss (Fig. 2). In addition to the mechanical stresses on the light-conducting glass core, the increase of noise as a function of LAF depends on the characteristics

of the glass core in the cable and how it is coupled to the outer cable (sheath, protection) and these are specific to the batch of the light-conducting glass core used during manufacturing of the cable. Nevertheless, the estimates obtained for instrument noise along the fibre can be used as a first order estimates when designing future DTS measurement campaigns with inevitable signal artifacts to the fibre-optic cable.

The value for $\sigma_{noise}$ at zero LAF is an intrinsic property of the measurement device and describes the noise level indepen-

dently of fibre type or the measurement conditions. The variability in the instument noise at zero LAF was evaluated using a similar fit as in Fig. 1 for each 30-min measurement period. The obtained zero LAF noise estimates varied between 0.50 K and 0.66 K (5th and 95th percentiles of the obtained values) during the measurement campaign; the variability was not fully ran-

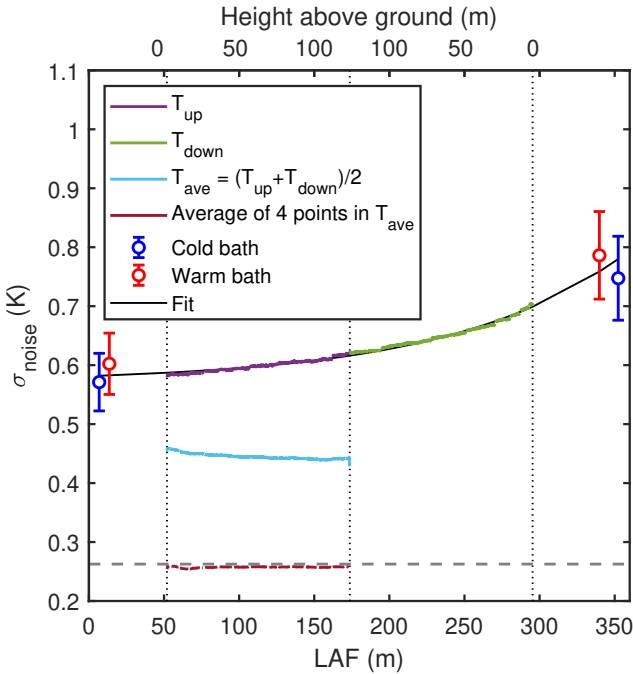

**Figure 1.** Average dependence of noise (1-$\sigma$) estimated with Eq. 10 for length along the fibre (LAF) and height above ground (top x-axis). Circles show $T$ variability in the calibration baths (mean $\pm$ SD). Dashed grey line shows the mean T standard deviation measured with 3D sonic anemometer at 27 m height during the measurement campaign. The vertical dotted black lines show the limits (bottom and top) of the vertical DTS measurements on the mast. The black line is a fit ($\sigma_{noise} = 0.57\text{K} + 0.01\text{e}^{8.6*10^{-3}\text{LAF}}$, $R^2$=0.99) to the noise estimates calculated from the cable going up ($T_{up}$) and down the mast ($T_{down}$). The noise level decreased after averaging the up and down portions of the measurements($T_{ave}$, cyan line) and fell below the typical sonic T variability level if further four spatially consecutive measurements were averaged.

dom as a systematic temporal component was also evident. Considering Eq. 4, the zero LAF noise in $T$ is likely related to the variability in laser intensity or detector sensitivity. The DTS instrument contains an internal reference coil of fibre-optic cable used for a first instrument-based calibration using the variability of the Stokes signal ($P_{s,in}$) from this reference coil. We used this as a joint proxy for both the variability in emitted light and detector sensitivity, since the signal has not yet experienced any

5   attenuation. The zero LAF noise displayed a linear dependence on this proxy ($y = (0.10 \pm 0.001\,\text{K/dB})\,x - (0.44 \pm 0.001)\,\text{K}$, $R^2$=0.87, where $y$ equals $\sigma_{noise}$ at LAF=0 m and $x$, decrease in $P_{s,in}$ (in dB) from its maximum value). Interestingly, the gradient is similar to that obtained for $\sigma_{noise}$ increases at step losses (Fig. 2) indicative of a more general dependence between signal intensity and $\sigma_{noise}$. It is known that laser intensity and detector responsivity can be sensitive to changes in temperature; the zero LAF noise did show a non-monotonic dependence on instrument internal temperature (calculated from data origi-

10   nating from the internal coil), and displayed additional changes when the instrument was cooling down or warming up (not





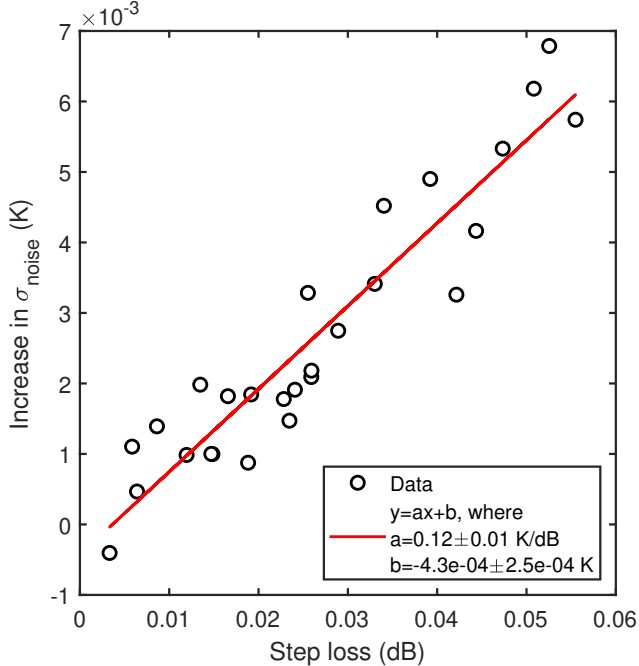

**Figure 2.** Increase of noise at step losses (i.e. sudden decreases of $P_s$ and $P_{as}$) along the cable versus step loss magnitude.

shown). The dependence was not fully explained by the instrument internal temperature alone and further studies are necessary to investigate this further.

## 3.2 High-frequency response of the DTS system

The high-frequency response of the DTS system was evaluated by comparing the power spectra of DTS measurements against

those of the co-located sonic anemometers as a reference (Fig. 3). The DTS power spectrum was dominated by white noise in the high frequency part of the spectrum, whereas the reference followed the canonical inertial subrange slope. Following the commonnly applied procedures for EC data processing (e.g. Ibrom et al. (2007)) the noise was removed from the DTS power spectrum by assuming that it was not correlated with the signal (see Sect. 2.2.1) and that the high frequency response could be estimated from the ratio between the noise-removed power spectrum and the EC reference (see Sect. 2.2). Comparing DTS

measurements to sonic anemometers at different heights (Table 1) provided slightly different values for the response time $\tau$ (Eq. 5), likely due to the uncertainty of the noise removal procedure. For reference, the values obtained are in the same range as found for water vapour EC flux measurements in high relative humidity conditions (Ibrom et al., 2007; Mammarella et al., 2009; Runkle et al., 2012), and are significantly higher than for carbon dioxide (Mammarella et al., 2009) or methane (Peltola et al., 2014) EC flux measurements. When comparing the DTS power spectra to EC power spectra it is important to recognise

that the instruments were sampling at different rates (DTS with 0.5 Hz and EC with 10 Hz) and hence the power spectra cannot

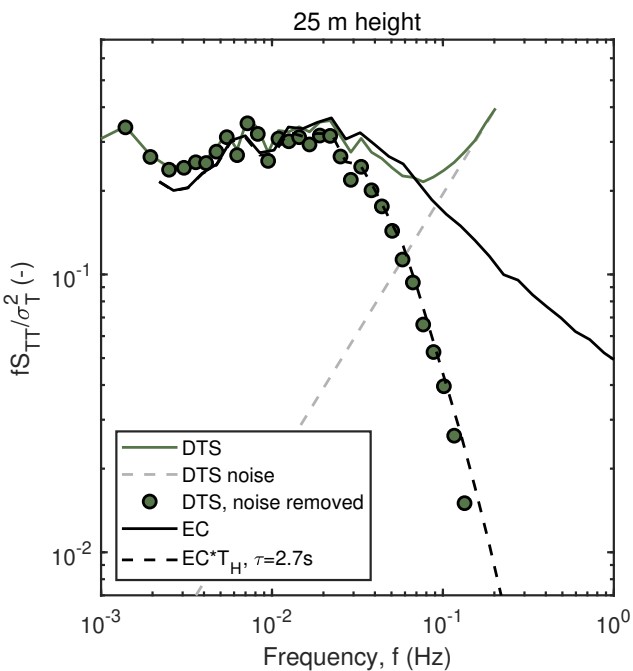

**Figure 3.** Ensemble-averaged, frequency weighted and normalised temperature power spectrum estimated from the DTS data and colocated 3D sonic anemometer (reference). Only periods with high DTS SNR and moderate wind speed (between 1.7 and 3.3 m s$^{-1}$) were selected for the ensemble. The DTS power spectrum is shown before and after the noise removal procedure and the high frequency attenuation is demonstrated by multiplying the reference power spectrum with a transfer function (Eq. 5).

fully be compared across all frequencies. The estimated response times and Eq. (5) should only be considered as a function for matching DTS and EC spectra, and not as a transfer function describing the functioning of the DTS system.

Theoretically, the high frequency response of the DTS system depends on 1) heat conduction across the quasi-laminar boundary layer surrounding the fibre cable, 2) heat conduction within the cable and 3) the spatial and temporal sampling

5  resolution of the DTS instrument (Thomas et al., 2012). The latter two can be approximated to be independent of environmental conditions, whereas the first one depends on the depth of the boundary layer surrounding the cable which is inversely dependent on wind speed. The DTS system high frequency response time $\tau$ was calculated for different wind speed conditions at different heights and no clear dependency on wind speed was found. This indicates that it is likely that the sampling resolution of the DTS instrument dominates the high frequency response and, hence, $\tau$ can be assumed to be constant for this particular fibre

10  cable and DTS instrument combination.

The attenuation of the temperature variance due to the limited high frequency response of the DTS system was estimated using the obtained values for $\tau$ and Eq. 6. The median values of the calculated $AF$ values for different heights ranged between 0.74 and 0.86 indicating that the DTS temperature variances were typically underestimated by 20–30 % (Table 1). The signal attenuation is, in effect, a combination of attenuation ($\tau$) and turbulent time scales (Horst, 1997), hence $AF$ was not constant





**Table 1.** Response times ($\tau$) describing the high frequency response of the DTS system derived though comparison with EC measurement systems at different heights above ground (cf. Eq. 5) and attenuation factors ($AF$) describing the attenuation of temperature variance due to the high frequency loss of the signal. 200 estimates for $\tau$ were estimated by bootstrap sampling the available data for each height and the reported values are the medians (interquartile range) of the 200 estimates. $AF$ was estimated for both height-dependent and constant $\tau$ for each 30-min time period during the measurement campaign with Eq. 6 and the reported values are the medians (interquartile range) of the $AF$ time series.

| Height above ground (m) | $\tau$ (s) | $AF$ (-) | $AF, \tau = 2.5\,s$ (-) |
|---|---|---|---|
| 5.5 | 2.9 (2.4...3.5) | 0.86 (0.80...0.91) | 0.88 (0.83...0.92) |
| 25 | 2.3 (2.1...2.4) | 0.78 (0.70...0.84) | 0.76 (0.69...0.83) |
| 27 | 3.1 (2.8...3.3) | 0.74 (0.67...0.82) | 0.76 (0.70...0.83) |
| 68 | 2.0 (1.6...2.3) | 0.78 (0.66...0.85) | 0.78 (0.66...0.85) |
| 126 | 1.4 (1.0...2.1) | 0.85 (0.73...0.90) | 0.81 (0.68...0.89) |

while $\tau$ was. Turbulent time scales were estimated using roughness sublayer scaling, $U/h_c$ (Thomas and Foken, 2007a). The turbulent time scales are modulated by atmospheric stability, hence $AF$ should also depend on stability. When calculated with fixed value for $\tau$ for each height, $AF$ showed both an approximately linear decrease with $U/h_c$ and a dependence on stability (Fig. 4). However, the estimates of $AF$ at different heights did not follow the same linear dependence, with clear differences

between the dependence at 68 m and 126 m (Figs. 4d and 4e) compared to lower heights, which may indicate that $h_c$ was not the physically meaningful length scale at these heights. The dependence of signal attenuation on turbulent time scales did cause the median $AF$ to increase weakly with height above the canopy indicating smaller signal attenuation well-above the canopy (Table 1). However, $AF$ values from different heights were typically within 5 % of each other, indicating that the whole profile was attenuated in a similar fashion and the reported values for $AF$ serve as first-order estimates for the signal attenuation

throughout the profile. Values of $AF$ were higher below the canopy, indicating a reduction in the high frequency attenuation of the signal; this was expected since organised (slow) motions have been previously found to dominate the variability within forests (Thomas and Foken, 2007a, b).

### 3.3 Second- and third-order statistics

Figure 5 and Table 2 show the agreement between temperature variances derived from DTS and the colocated 3D sonic

anemometers. The DTS temperature variance was estimated using Eq. 9, where the noise variance has been compensated for prior to the comparison. The temperature variance was dominated by large coherent eddies connected to the low-frequency end of the spectrum. Hence, the DTS measurements were able to capture the variability of this most important canopy flow mode accurately, in other words, the bulk of the time series variance was related to fluctuations close to the peak of the power spectra and DTS system was able to resolve the variability at these frequencies (Fig. 3). Note that the gradient of the linear fits

between the temperature variance estimates (Table 2) were close to the values obtained for the attenuation factors describing





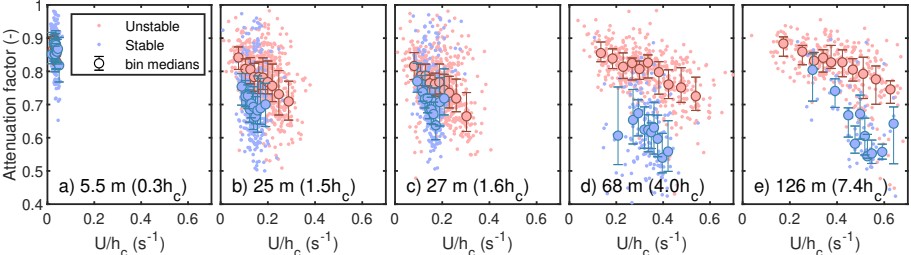

**Figure 4.** Dependence of $AF$ on $U/h_c$ at different mast heights. 30-min values (points), bin medians (filled circles) and the bin interquartile range (error bars) are shown. Data were screened based on flux stationarity (Foken and Wichura, 1996) and values for temperature variance and $\overline{w'T'}$ before plotting ($|\sigma_T^2| > 0.02\mathrm{K}^2$ and $|\overline{w'T'}| > 0.015\mathrm{Km/s}$).

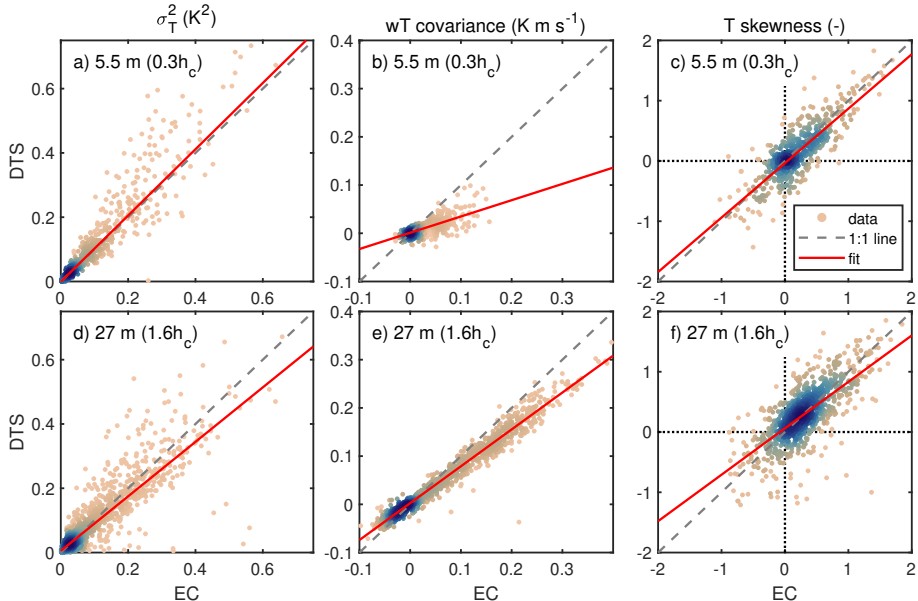

**Figure 5.** Comparison of DTS and EC 30-min values for temperature variance (left column), $\overline{w'T'}$ covariance (centre column) and skewness (right column) at two different heights on the mast. Upper panels a)–c) show below-canopy ($0.3h_c$) and lower panels d)–f) show above-canopy ($1.6h_c$). For skewness, only periods with SNR>0.5 (Eq. 13) were used, otherwise all available data was included. Coefficients related to the linear fits are given in Table 2. Colour denotes the density of the point cloud. Note that no corrections for sensor separation effects (Horst and Lenschow, 2009) or signal attenuation (Sect. 3.2) were made prior to comparison.

the high frequency loss (Sect. 3.2) suggesting that the systematic mismatch between DTS and EC temperature variances was solely related to the limited high frequency response of the DTS system.

Following Thomas et al. (2012), the kinematic heat fluxes (i.e. $\overline{w'T'}$) were calculated using the vertical wind speed from 3D sonic anemometer and temperature either from the same 3D sonic anemometer or colocated DTS signal. Above the canopy,
5 heat fluxes calculated from both systems showed relatively good agreement (Fig. 5e), indicating that the DTS system was





**Table 2.** Linear regression statistics between DTS and 3D sonic anemometers at different heights (y = ax+b, where y equals DTS and x 3D sonic anemometer). The fits between datasets were performed using robust regression in order to minimise the effect of outliers on the regression coefficients. Standard errors for the coefficients are given in parentheses and N shows the number of 30-min data points used in the fit. For skewness, only periods when the signal to noise ratio for DTS data was above 0.5 were utilised, for other statistics all available data were used. Note that no corrections for sensor separation effects (Horst and Lenschow, 2009) or signal attenuation (Sect. 3.2) were made prior to comparison.

| Height (m) | Variance | | | Covariance, $\overline{w'T'}$ | | | Skewness | | |
| --- | --- | --- | --- | --- | --- | --- | --- | --- | --- |
| | a (-) | b ($K^2$) | N | a (-) | b (Km/s) | N | a (-) | b (-) | N |
| 5.5 | 1.03 (0.003) | <0.001 | 886 | 0.34 (0.005) | 0.001 (0.001) | 894 | 0.90 (0.035) | -0.036 (0.016) | 505 |
| 25 | 0.80 (0.004) | <0.001 | 1336 | 0.60 (0.005) | 0.001 (0.001) | 1353 | 0.92 (0.041) | 0.036 (0.019) | 885 |
| 27 | 0.85 (0.005) | 0.006 (0.001) | 1333 | 0.76 (0.003) | 0.003 (0.001) | 1353 | 0.77 (0.071) | 0.062 (0.037) | 853 |
| 68 | 0.87 (0.001) | <0.001 | 1332 | 0.73 (0.004) | 0.002 (0.001) | 1353 | 0.86 (0.036) | 0.010 (0.025) | 720 |
| 126 | 0.90 (0.007) | <0.001 | 1325 | 0.83 (0.003) | <0.001) | 1353 | 0.76 (0.143) | 0.043 (0.087) | 651 |

able to resolve most of the eddies contributing to the vertical turbulent flux. However, within the forest canopy the agreement was worse (Table 2 and Fig. 5b). There are at least two reasons for this: 1) within the forest heat transport is dominated by small eddies since the coherency of large eddies typically dominating surface layer heat transfer is broken up by the canopy elements (spectral short circuit; Finnigan (2000); Launiainen et al. (2007)) and forest floor is close which also suppresses the

size eddies dominating heat transfer and DTS cannot capture the small scale turbulence; and 2) there was a 3.5 m horizontal sensor separation between the 3D sonic anemometers and DTS fibre cable which significantly contributed to dampening the high-frequency response of the joint anemometer-DTS flux calculation, especially close to the ground (Horst and Lenschow, 2009). This is supported by the finding that the in-canopy temperature variance was captured accurately, while the heat flux was not, meaning that the temperature fluctuations related to the large eddies sweeping into the in-canopy were captured accurately

yet their signal was decorrelated with respect to the vertical wind speed (i.e. vertical turbulent flux) by the canopy elements. This is evident, for instance, in the comparison between within and above forest canopy power spectra in Launiainen et al. (2007) measured at the same site. In general, scalars (e.g. temperature) behave very differently to the vectors within the canopy (Vickers and Thomas, 2014).

Third-order statistics such as temperature skewness were also compared. Again, the effect of noise was removed following

the method by Lenschow et al. (2000) and the temperature skewness values were calculated using Eq. 11. Additionally, all data with SNR<0.5 was discarded for the comparison presented in Fig. 5 and Table 2. The reason for this additional filtering can be seen in Fig. 6, where the discrepancy between DTS and EC temperature skewness increases rapidly as DTS SNR decreases below 0.5. The dependence of DTS SNR on height meant that better estimates of temperature skewness were available closer to the canopy than at higher elevations above. Imposing a DTS SNR threshold of 0.5 still left 54 % of all DTS data measured

during the campaign and 63 % of DTS data measured below 50 m height available for generating third-order statistics, hence





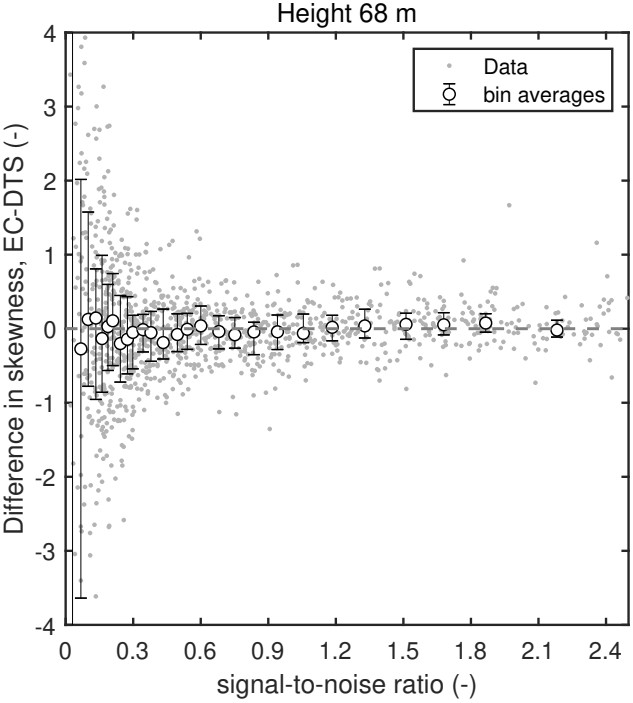

**Figure 6.** Difference in temperature skewness calculated from 3D sonic anemometer and DTS as a function of DTS signal-to-noise ratio (Eq. 13). Individual 30-min values (grey dots), bin medians (circles) and the interquartile range (error bars) are shown.

we conclude that this DTS system can still be used after SNR filtering to monitor higher-order statistics and the non-Gaussian character of the flow despite the higher noise floor of this instrument relative to the older 2 km variant used in Thomas et al. (2012).

### 3.4 Profiles of first- through third-order statistics

5    A comparison of the profiles of turbulence statistics in different stability regimes are shown in Fig. 7. DTS provides continuous profiles, whereas conventional measurements provided point estimates; this is the unique strength of the spatially continuous DTS measurements. The mean potential temperature gradients from DTS but exhibited some bias when compared against reference instrumentation (Fig. 7a), especially in mildly stable and strongly unstable cases. The fibre-optic cable was not radiation-shielded and hence it was exposed to shortwave and longwave radiation transfer with its surroundings. 30-min mean

10    temperature values were found to be biased by solar heating during daytime and longwave radiative cooling at night in accordance with prior studies (de Jong et al., 2015; Sigmund et al., 2017). Due to the forest canopy, these radiation-induced biases differ below and above the canopy, introducing biases into the temperature gradient. In general, during night, DTS showed a greater decrease in temperature with height within the canopy than the reference profile and the bias in the gradient depended on the amount of longwave radiative cooling (not shown). During daytime, DTS showed a weaker decrease in temperature





with height due to canopy shading; this was particularly evident during morning and evening but less during the middle of the day. The biases in across-canopy temperature differences were 0.07 K (median bias in temperature difference between 27 m and 5.8 m heights) at night and 0.04 K at daytime, indicating that the DTS profile overestimated the across-canopy temperature gradient at night and underestimated the gradient during daytime. For comparison, the unbiased median temperature gradients

during these periods calculated from the reference measurements were 0.34 K and -0.29 K at night and daytime, respectively.

As shown above, DTS measurements and 3D sonic anemometers showed good agreement in different mixing conditions within the canopy sublayer, slightly above the canopy and well above the forest. As expected, the temperature variance peaked at the canopy top due to 1) strong turbulence production originating from wind shear, and 2) canopy heat source/sink related to solar heating (daytime) or radiative cooling (night time) of the canopy. The scaled temperature variability ($\sigma_T/|T_*|$, where $T_*$

was calculated based on Eq. 3 and data from 27 m height) followed Monin-Obukhov (M-O) similarity scaling in the unstable and near-neutral regime above about 50 m height, indicating that surface layer scaling was valid between this height and the mast top. As expected, at lower heights (between 50 m height and ground floor), the measured values for $\sigma_T/|T_*|$ departed from the M-O predictions. This is typical for the roughness sublayer, where the turbulence resembles more mixing layer than boundary layer turbulence (Raupach et al., 1996; Finnigan, 2000; FINNIGAN et al., 2009) meaning that the turbulent mixing

is more efficient than in the surface layer and hence a smaller $\sigma_T$ will already yield the same turbulent flux. In other words, the correlation between $w$ and $T$ is higher in the roughness sublayer than in the surface layer (Patton et al., 2010) and hence M-O scaling is no longer valid. These results suggest that the height of the roughness sublayer at this sites is about 50 m, which corresponds to approximately three times the height of the roughness elements (i.e. trees). This is in line with previous study at this site (Rannik, 1998) and other studies made in wind tunnels or at other forest sites, where estimates for the roughness

sublayer height typically range between $2h_c$ to $5h_c$ ($3h_c$ being the most common estimate) where $h_c$ is the height of the roughness elements (Garratt, 1980; Coppin et al., 1986; Mölder et al., 1999; Poggi et al., 2004; Thomas et al., 2006).

In near-neutral situations the scaled temperature variability exceeded the predictions made with M-O scaling, since the heat fluxes (and hence also $|T_*|$) decreased with $\zeta$ yet the temperature variability ($\sigma_T$) did not decrease at the same rate; i.e. heat transfer efficiency approached zero at the neutral limit (e.g. Rannik, 1998). Under stable stratification the scaled temperature

variability showed consistent z-dependence between the displacement height and approximately 30 m height, whereas above 30 m it was relatively independent of height following z-less stratification and any variability above this height was most likely related to flow processes other than interaction with the surface. These observations are in line with previous findings (e.g. Pahlow et al., 2001). Note that here, $|T_*|$ was calculated using measurements at a fixed height (27 m), i.e. local scaling was not utilised. The profiles shown in Fig. 7b for unstable and neutral periods could be expected to show similar pattern even if

local scaling would be used since turbulent fluxes can be conjectured to be constant with height in the bottom part of the ABL. However, for stable situations the $\sigma_T/|T_*|$ calculated based on local scaling might depart from what is shown here since $|T_*|$ varies with z.

Temperature skewness was alomst independent of height (approx. 0.5–0.6) in unstable conditions above 30–50 m ($2h_c$–$3h_c$) which corresponded to the roughness sublayer height found above. Skewness increased as the instability increased. Skewness

decreased when moving to the canopy sublayer, but stayed positive throughout the profile, indicating non-Gaussian flow. Non-

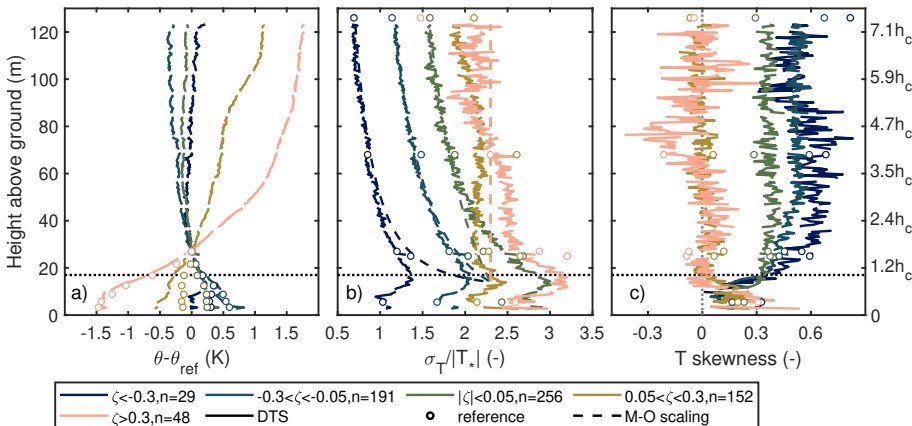

**Figure 7.** Profiles of potential temperature gradient ($\theta_{\mathrm{ref}}$=potential temperature at 27 m), normalised temperature variability ($\sigma_T/|T_*|$) and temperature skewness derived from DTS (continuous lines) and reference (dots) measurements. The data were binned based on the stability parameter $\zeta$ estimated from EC measurements at 27 m height. $\sigma_T/|T_*|$ was also calculated based on Monin-Obukhov similarity theory (dashed lines) for unstable stratification using $\sigma_T/|T_*| = 2.3\,(1+9.5|\zeta|)^{-1/3}$ (Cava et al., 2008). Note that local scaling was not utilised, i.e. $|T_*|$ was calculated using measurements at a fixed height (27 m). Only periods with high SNR were used and n denotes the number of 30-min periods in each stability bin. The right-hand axis displays height as a fraction of canopy height ($h_c$) and the canopy height is denoted with a horizontal dashed line.

zero skewness is often connected to an imbalance between the transport of scalar in question by ejective and sweeping air motions (Katul et al., 2018). Hence, based on these results the ejections and sweeps were not in balance and the ejective motions dominate heat transport in the unstable conditions throughout the air column. Non-zero skewness of scalar time series has been also linked to the influence of atmospheric boundary layer (ABL) scale large eddies entraining air from the free

troposphere and transporting it in downdrafts close to the ground (Mahrt, 1991; Couvreux et al., 2007; van de Boer et al., 2014). The measured skewness profiles are qualitatively similar to those from the wind tunnel study of Coppin et al. (1986) and large eddy simulations (LES) of Patton et al. (2015) obtained with canopy-resolving LES coupled with a multi-level canopy model. In contrast to the profiles in unstable conditions, in stable conditions above the forest canopy the flow was Gaussian (zero skewness), whereas skewness was positive below the canopy due to the sweeping motions penetrating through the canopy

and bringing pulses of warm air from aloft into the cold below-canopy air.

### 3.5 Examples of organised flow patterns observed with the DTS system across vertical coupling regimes

The spatially-continuous measurements of the DTS system enabled the detection and analysis of spatial patterns in the flow. Figure 8 shows a measurement example during a daytime unstable regime. Large coherent eddies dominated the variability (Fig. 8b) and their detailed vertical structure was captured with the DTS measurements. Positive temperature perturbations

were correlated with upward vertical air motions and negative with downward motions, indicating upward-directed sensible heat flux due to the unstable stratification. The entire 120 m measurement domain was coupled due to large coherent eddies





effectively mixing the air throughout the vertical. The temporal extent and amplitude of the temperature fluctuations changed with height, corresponding to changes in the dominant eddy size and flux magnitude with height. The patterns with low temperatures (downward air motions called sweeps) occasionally reached the forest floor, but did not always penetrate the whole canopy layer (between 10 and 17 m) since they were likely disrupted by the canopy passage. The patterns with high

temperatures representing upward motions called ejections typically originated from the forest floor, traversed the forest canopy and continued moving upward (compare Fig. 8a and 8b). However, note that these analyses rely on Taylor's frozen turbulence hypothesis, i.e. on the assumption that the two dimensional spatial details of the coherent motions can be delineated from the vertical DTS measurements. Ramp-cliff patterns connected to the ejection-sweep cycle were evident in the time series collected just above the canopy (Figs. 8e and 8f), but not so clearly at higher levels or below the canopy (Figs. 8c, 8d and 8g).

These patterns have been previously suggested to be connected to the wind shear and corresponding inflection point instability close to the canopy top (Raupach et al., 1996; Finnigan, 2000; Cava et al., 2004; Thomas and Foken, 2007b; Göckede et al., 2007). For this reason the signatures of ejections would not be expected to reach very high above the canopy. Mean potential temperature profile (Fig. 9a) showed unstable stratification, except below canopy and in the upper parts of the profile where the stratification was close to neutral.

The patterns observed with the DTS system and the 3D sonic anemometer at the mast top agreed qualitatively with the vertical-pointing windlidar (Fig. 8), yet quantitative analysis on the agreement was hindered by the fact that the lidar instrument was located approximately 400 m upwind from the measurement mast. While the temporal lag caused by this horizontal displacement was taken into account by shifting the lidar time series in time to maximise the correlation with the mast-top 3D sonic anemometer, the eddies may have already been deformed during advection from the lidar measurement location to

the measurement mast, in addition to changes in wind direction. Hence, a direct comparison between the lidar in its current position and the mast measurements is not possible. Nevertheless, these observations do demonstrate the capabilities of joint measurements with lidar and DTS in capturing continuous vertical profile of turbulent motions from forest floor up to boundary layer top.

In contrast to the daytime example, during the night time example the air temperature patterns suggested a decoupling caused

by the strong thermal stratification of the air (Fig. 9a and 10b) resulting from radiative cooling of the surface (net radiative loss of 51 W/m$^2$) and low mechanical production of turbulence (i.e. weak wind shear). Two strong temperature inversion layers were evident: one at the canopy top between 12 m and 25 m heights and one at 80 m which descended down to 60 m height during the latter part of the period (Fig. 11a). Vertical movement of these inversion layers resulted in a bimodal vertical profile for $\sigma_T$ during this period (Fig. 9b). These inversion layers decoupled the air column into three layers: below canopy

air space (below 10 m height), canopy layer with organised motions (between 10 and 80 m) and a residual layer with slowly varying non-turbulent motions (above 80 m). The maximum potential temperature gradient resided at the canopy height due to radiative cooling of the canopy (Fig. 9a and 11a). Flow patterns were evident, in particular close to the canopy top. Inverted ramp patterns were observed in the temperature time series at the beginning of the example period (Fig. 10) and a canopy wave was observed in the middle of the period (Fig. 11a). By following the evolution of the maximum temperature gradient (Fig.


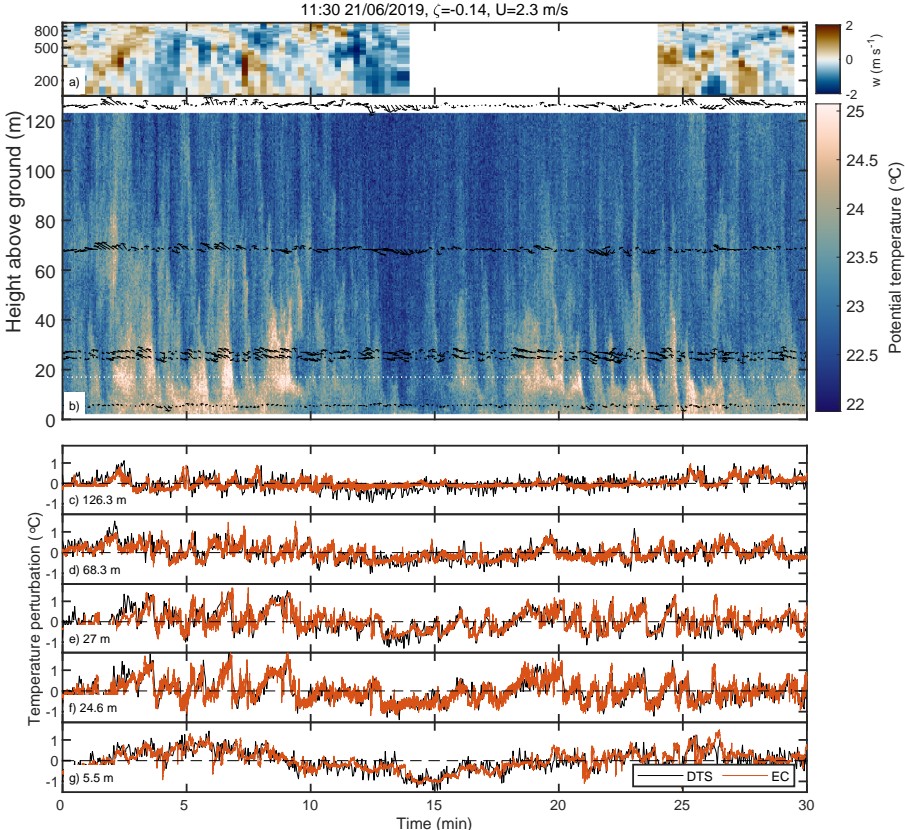

**Figure 8.** Example of simultaneous lidar, DTS and 3D sonic anemometer measurements during daytime convective conditions. Panel a): lidar vertical wind velocity component (gap is due to non-vertical operation during scanning sequence). Panel b): temperature profiles measured with the DTS system (color) and wind speed fluctuations (arrows) measured with 3D sonic anemometers. For illustration purposes the wind speed fluctuations were averaged over 10 s and the vertical wind component was multiplied by 10 prior to plotting. Length of the arrows denote the magnitude of the wind speed fluctuations and direction is determined by the sign of vertical ($w'$) and horizontal ($u'$) wind speed fluctuations (up: $w' > 0$, down: $w' < 0$, left: $u' < 0$, right: $u' > 0$). Panels c)-g): temperature perturbations ($T - \overline{T}$) measured with DTS and sonic anemometers at different heights on the mast. Local time (UTC+2), the corresponding stability parameter ($\zeta$) and mean wind speed (U) measured at 27 m height are given in the figure title. Canopy height was approximately 17 m and is highlighted with a white dotted line in panel b). For illustration purposes the data gaps from cable holder locations were filled with linear interpolation prior to plotting.

11a), the amplitude of the canopy wave was estimated to be about 8 m. The switch from inverted ramps to canopy waves was likely due to a decrease in wind shear and, hence, a decrease in turbulence production (Fig. 11b).

These two examples from contrasting mixing regimes illustrate the unique observational capabilities of the DTS to analyse the spatial structure of the organised flow patterns, which can only be guessed at individual heights from classical sonic anemometry (compare e.g. Figs. 8b and 8e). In principle similar results could have been obtained with an array of several in-





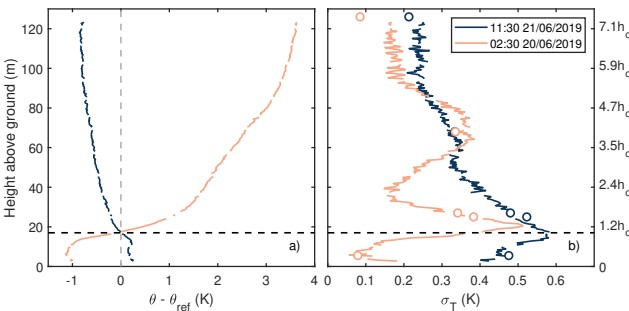

**Figure 9.** Mean potential temperature gradient (subplot a)) and standard deviation (subplot b)) profiles for the two example periods shown in Figs. 8 and 10. Dots in subplot b) show the $\sigma_T$ values estimated with the 3D sonic anemometers at different heights.

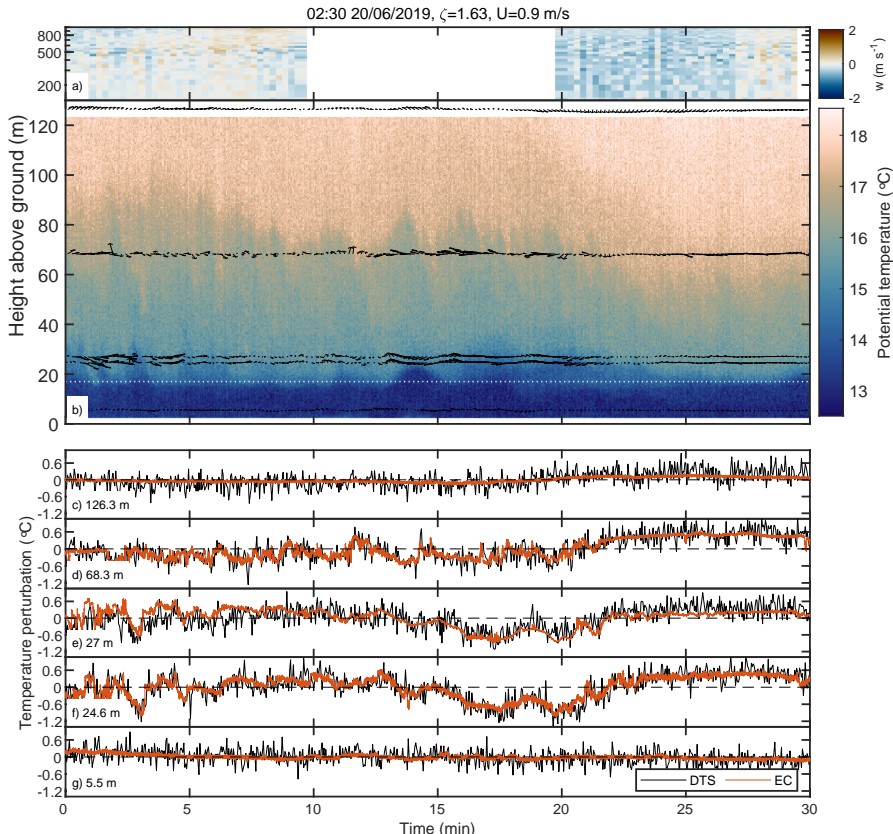

**Figure 10.** Same as Fig. 8 but for a strongly stable situation at night.

situ instruments (Gao et al., 1989; Lee et al., 1997; Poulos et al., 2002; Horst et al., 2004; Mahrt and Vickers, 2005; Mahrt et al., 2009; Patton et al., 2010; BOU-ZEID et al., 2010; Feigenwinter et al., 2010; Thomas, 2011; Serafimovich et al., 2011; Mahrt et al., 2014), yet such measurement setups do not provide spatially continuous measurements with the same spatial resolution

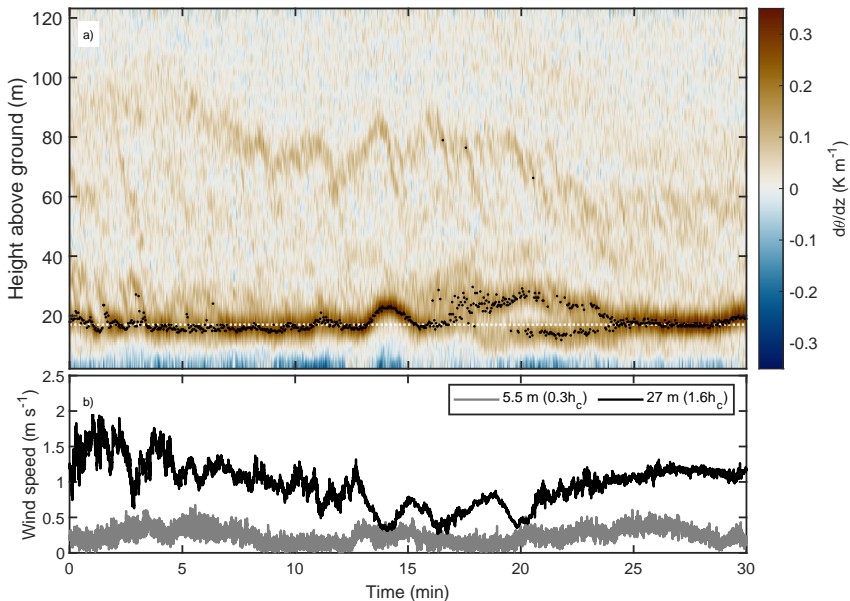

**Figure 11.** Evolution of potential temperature gradient (d$\theta$/dz) (subplot a) for the period shown in Fig. 10 and concurrent wind speed time series from two heights (subplot b). The black dots in subplot a) highlight the location of the maximum gradient at each time step. The gradient was calculated using spline fit for each time step and estimating the gradient for each height from the first derivative of the fit. White dotted line shows the canopy height.

as the DTS system and are labour-intensive and require careful cross-instrument calibration. Furthermore, continuous profiles are required for following the evolution of e.g. inversion layers (see Fig. 11) and hence not easily detectable with arrays of in-situ instruments.

## 4  Conclusions and outlook

5  This study demonstrated the unique observational potential of DTS measurements to capture the salient features of atmospheric flow within and above forest canopies across contrasting flow regimes. Despite the fact that these results were obtained with a different DTS machine compared to Thomas et al. (2012) and that different cable suspension techniques were used, these results are in line with, and complement their findings. In spite of the higher noise floor in the measurements and the limited high-frequency response compared to sonic anemometers, we found the DTS technique to accurately capture the second- and

10  third-order moments, which allows for detection of the spatial structure of coherent air motions dominating the temperature variability at the forest-air interface. In accordance with Thomas et al. (2012), the measurements performed best when either large variability (high heat fluxes) or strong stratification (low mixing and clear skies) was present. Therefore, DTS mea-





surements can provide the missing spatial details of atmospheric mixing close to the surface which cannot be acquired with conventional in-situ or remote sensing methods.

A combination of DTS and conventional in-situ instrumentation has been shown to be advantageous when studying low-mixing nocturnal air flows (Thomas et al., 2012; Zeeman et al., 2015; Pfister et al., 2017; Mahrt et al., 2019; Pfister et al., 2019)
and morning transitions (Higgins et al., 2018) over smooth surfaces. In this study it was shown that the DTS measurements can be used in a wide range of mixing conditions throughout the day, both above and within rough forest canopies. When affixed to a tall mast, DTS measurements can capture coherent motions up to heights of $7h_c$ (Fig. 8b) or more. Hence, observations with different DTS measurement configurations should prove useful in locations where the flow exhibits persistent spatial patterns which are not transported with the mean flow and, thus, cannot be studied with standard point-based in-situ instrumentation.
These locations include, but are not limited to, flows within and just above forest canopies and street canyons, and flows over edges and other spatial discontinuities in the surface characteristics.

Together with aiding the analysis of different flow modes, DTS measurements show potential for investigating the spatial details of atmosphere-ecosystem interactions and gas transfer between these two domains. For instance, the vertical profile of temperature fluctuations can help separate the canopy and forest floor contributions to above-canopy scalar fluxes (Thomas
et al., 2008; Klosterhalfen et al., 2019) since the measurements allow tracking the temperature signal related to coherent ejection and sweep air motions through the forest canopy (Fig. 8b). The measurements can also aid in trying to decipher the persistent problem related to advection at the EC flux measurement sites when trying to close the mass balance of e.g. $CO_2$ for a given target ecosystem (Aubinet et al., 2010), since part of this issue is likely to be related to the unresolved horizontal variability of turbulent mixing within the forest (Feigenwinter et al., 2010).
Ultimately the DTS measurements enable studies to go beyond the typical time series from isolated point-measurements, yet at the same time provide measurements at the same temporal and spatial scale as conventional in-situ instruments. Many topical open questions in the field of ecosystem-atmosphere interactions require spatio-temporal information on the processes involved, and DTS has been shown here to be a suitable tool for gathering such information.

*Data availability.*  Data has been uploaded to open data repository Zenodo (Peltola et al., 2020).

*Author contributions.*  OP designed the experiment and did the data processing and analysis. KL and CKT supported the DTS data analysis work. IM provided technical support in the field. EO maintained the Doppler lidar instrument and provided the data. OP wrote the first version of the manuscript and all authors provided input.

*Competing interests.*  The authors declare that they have no conflict of interest.



*Acknowledgements.* Technical staff at the Hyytiälä research station is acknowledged for their help during the measurement campaign. OP is supported by the postdoctoral researcher project (decision 315424) funded by the Academy of Finland. KL and CT received funding from the European Research Council (ERC) under the European Union's Horizon 2020 research and innovation program (grant agreement No 724629, project DarkMix). ICOS-Finland and Atmospheric Mathematics (AtMath) projects by University of Helsinki are also acknowledged.



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
