# Peer review of "Suitability of fiber-optic distributed temperature sensing to reveal mixing processes and higher-order moments at the forest-air interface"

_Atmospheric Measurement Techniques, 2020_

## Referee Comment (RC1) · Anonymous Referee #3 · 12 Oct 2020

Review of Suitability of fiber-optic distributed temperature sensing to reveal mixing processes and higher-order moments at the forest-air interface by Peltola et al.

General Comments

Overall, the manuscript presents the usefulness of a new and increasingly important tool with respect to spatial measurements of turbulence and atmosphere-ecosphere interactions. The manuscript is generally well put together but are a few places where some clarification is needed (see minor/technical comments below) to improve the message being presented. It is an interesting concept that adds to the toolbox of measurement techniques beyond the traditional flux tower setup to investigate the underlying turbulence that drives scalar fluxes in different landscapes and glad to see if developed further.

Across the manuscript, the spatial extent/direction of the measurement needs to be clear. My only broader comment is in relation to the connection between the flow and temperature variability; the connection between these two values needs to be better supported; there are places where it appears the two variables are used interchangeably and it can lead to a little confusion for the reader (Sections 3.4). This is primarily an organizational issue, not a science issue.

Minor/Technical Comments

Pg2ln6: "practical point-of-view" meaning a sufficient amount of spatially distributed tower measurements?

Pg2ln6: "Hence..." Not sure how many spatial stats are being derived directly from time series observations at a single point; this needs to be clarified; its point is a little convoluted. I will agree that eddy structure, particularly in the vertical, is assumed from time series using Taylor's hypothesis but I don't necessarily agree that spatial statistics are being derived through Taylor's hypothesis.

Pg2Ln8: "Another motivation..." feels tacked on; could be better framed to fit within the overall context of the need for more spatially explicit measurements.

Pg3Ln23: "Furthermore, we evaluate..." remove "also" from this sentence.

Pg3Ln25: "Deviations from Gaussian distribution are..."; assuming distribution of temperature? Clarify.

Pg5Ln19-23: "When comparing the results..." Not sure these two sentences are needed; there isn't that strong of a comparison back to the Thomas et al., 2012 paper within the manuscript. Though this work is based off the Thomas et al., work, unless a direct comparison within this paper is going to be made, the differences in the instrument variant is does not need to be explicitly stated.

Pg6Ln16-19: "After determining the differential attenuation. . ." I suggest splitting this into two sentences at the semicolon instead of keeping it as one longer, complex sentence since it is detailing two distinct processing steps.

Pg6Ln23: "After quality filtering. . ." Out of how many potential 30-minute periods, either as a percent or total measurement periods?

Figure 3: Put the height of the sonic anemometer within the figure caption; easy to miss being the title of the graph.

Pg12Ln16: "The temperature variance was dominated. . ." This feels out of place; there isn't any direct support for this comment near this sentence and think it should be later in the paragraph/section after the discussion of Figure 5/Table 2 or in the previous section with the discussion of the power spectra comparisons. Also strikes me as being a statement that applies to above the canopy and not as much within the canopy due to the general size of eddies closer to the ground and surrounded by obstacles (trees).

Pg14Ln5: "size eddies dominating. . ." should be "size of eddies dominating. . ."

Pg15Ln7: Remove the "but" from "The mean potential temperature gradients from DTS bug exhibited. . .".

Pg16Ln4: "For comparison, the unbiased median temperature gradients. . .", Gradient between which heights; across canopy?

Pg16Ln15: Remove "already" in "hence a smaller. . ." sentence.

Pg16Ln33: The word "almost is misspelled.

Pg17Ln6-8: "The measured skewness profiles. . ." There is little to no context for this sentence as it requires the reader to look up these papers or be very familiar with them to understand the similarities in the skewness profiles. Clarify or add context to the sentence or remove.

Pg18Ln8-9: It might be better to mention of the caveat using Taylor's frozen turbulence

hypothesis on this analysis at the beginning of the section instead of in the middle of the paragraph, or break the paragraph in half to be clear which pieces of the analyses are relying on Taylor's hypothesis.

Pg18Ln32: "Flow patterns..." This sentence is not saying anything new that the next sentence is not already saying and is not a very useful sentence. Maybe combine the two.

---

## Referee Comment (RC2) · Anonymous Referee #2 · 18 Oct 2020

The manuscript entitled "Suitability of fiber-optic distributed temperature sensing to reveal mixing processes and higher-order moments at the forest-air interface" by Peltola et al. discusses the applicability of distributed temperature sensing (DTS) at a forest site for evaluating the mixing and higher-order moments across the forest canopy and above the vegetation. The distributed temperature sensing technique is a very interesting, relatively new, approach to get more insights into the mentioned topics, consequently, this paper is definitely answering relevant scientific questions within the scope of AMT. The experiment description is sufficient, methods and assumptions are valid and clearly outlined and the paper comes to substantial conclusions. The title reflects the content and the abstract gives a concise and complete summary of the

work presented. The paper is clearly structured and well written, hence, it is possible to follow the drawn conclusions. Before this work will be published, however, I would like to ask the authors to address the following list of specific comments on their work:

Abstract, L6: replace "quantified" with "assessed".

P4L26: what is the range of the Lidar? From which height until which height it is giving measurements?

P7section2.2.1: the noise is quite high, I would say. Frequencies of noise and turbulence can overlap; how do you separate the signals then? Elaborate this maybe a bit more here in the manuscript to give proof to the reader that the procedure you are applying yields the presented results.

P12L11 and P14L3: at the first place you are stating that organized slow motions are dominating within the canopy, at the second place you are saying that small eddies are dominating since the coherency is broken up: this is contradictory, please revise.

Fig6 and other places: you are using the temperature from the sonic anemometers...I guess you converted sonic temperature in "real" temperature before use, state this somewhere. General question to the use of temperatures derived via sonic anemometer: they are not very accurate when it comes to absolute numbers, how did you deal with this? For prospective studies it might be more feasible to use profiles of e.g. thermocouples for comparison with the DTS. I also wonder about the Gill sonic anemometer: did you have any issues with this sonic regarding noise? I heard from several cases that the Gill sonic type you used has a problem with noise, also have personal experience with that issue. How did you deal with noise in the sonic data, if you observed it?

Fig7: reformulate the figure caption in such way that it gets very clear that you applied the Cava parameterization only for unstable conditions: i.e., "For unstable conditions, ...was also calculated based on Monin-Obukhov......".

Fig8: the arrows are very hard to identify.

P17: regarding vertical coupling: I would love to see here a comparison between the conclusions regarding coupling/decoupling which you derive with DTS and other approaches, e.g. the correlation of standard deviation of vertical wind (cf. Thomas et al., 2013). This does not have to be a complete analysis, this is beyond the scope of your paper, but a figure comparing coupling/decoupling conclusions derived via DTS and the mentioned correlation of sigma(w) would be very valuable, also for prospective work in this area. You have the required instrumentation there, a couple of 3-D sonics in profile.

Conclusions: you touch very briefly the topic advection there; can you write to this a bit more? A vertical profile would not be sufficient for approaching advection issues, but a 3-D array could. How would such a setup in your opinion have to look like to approach advection sufficiently?

―――――――――――――――

---

## Author Comment (AC1) · 1 Dec 2020

Manuscript: Title: Suitability of fiber-optic distributed temperature sensing to reveal mixing processes and higher-order moments at the forest-air interface

Author(s): Olli Peltola et al.

MS No.: amt-2020-260

MS type: Research article

We thank both referees for their comments. Please find below point-by-point responses to the presented critique. Referee comments are in **bold**, responses with red and changes to the manuscript with blue.

**Anonymous Referee #2**

**The manuscript entitled "Suitability of fiber-optic distributed temperature sensing to reveal mixing processes and higher-order moments at the forest-air interface" by Peltola et al. discusses the applicability of distributed temperature sensing (DTS) at a forest site for evaluating the mixing and higher-order moments across the forest canopy and above the vegetation. The distributed temperature sensing technique is a very interesting, relatively new, approach to get more insights into the mentioned topics, consequently, this paper is definitely answering relevant scientific questions within the scope of AMT. The experiment description is sufficient, methods and assumptions are valid and clearly outlined and the paper comes to substantial conclusions. The title reflects the content and the abstract gives a concise and complete summary of the work presented. The paper is clearly structured and well written, hence, it is possible to follow the drawn conclusions. Before this work will be published, however, I would like to ask the authors to address the following list of specific comments on their work**

RESPONSE: We thank the referee for this positive comment.

**Abstract, L6: replace "quantified" with "assessed".**

CHANGES TO THE MANUSCRIPT: Done

**P4L26: what is the range of the Lidar? From which height until which height it is giving measurements?**

RESPONSE: The Doppler lidar provides data from 75 m to 9585 m in range with 30 m spatial resolution and the clearest signal typically originates from the boundary layer where the aerosol loading is the highest. The instrument is situated on the roof of a 5 m high building so that the first measurement height is at 80 m above ground.

**P7section2.2.1: the noise is quite high, I would say. Frequencies of noise and turbulence can overlap; how do you separate the signals then? Elaborate this maybe a bit more here in the manuscript to give proof to the reader that the procedure you are applying yields the presented results.**

RESPONSE: The noise level can indeed be quite high when compared to the signal. The technique to separate signal variance from noise variance is based on the assumption that the noise is white and uncorrelated with the signal. In such case power spectra of noisy measurements contain two

independent components (see Fig. 3 in the manuscript) and noise contributes to the autocovariance only at lag zero, which enable the separation of the effect of these two components (noise and signal) on different statistics (variance, skewness). Ability of the technique to estimate the noise contribution decreases as signal-to-noise ratio decreases (see Fig. 6 in the manuscript). Based on referee suggestion we will add a note about prior studies utilizing the technique, but opt not to dwell on the issue since the technique is already discussed and validated in several other papers (Langford et al., 2015; Mauder et al., 2013; Nakai et al., 2020; Nemitz et al., 2018; Peltola et al., 2014; Rannik et al., 2016). Hence addition of short text and reference to some of the prior studies should suffice.

CHANGES TO THE MANUSCRIPT: Added the following sentence on p7l13: "The method has been shown to reliably estimate the influence of noise on second- and third-order statistics in turbulence measurements in prior studies (e.g. Lenschow et al., 2000; Rannik et al., 2016; Nakai et al., 2020)."

**P12L11 and P14L3: at the first place you are stating that organized slow motions are dominating within the canopy, at the second place you are saying that small eddies are dominating since the coherency is broken up: this is contradictory, please revise.**

RESPONSE: Thanks for noting this inconsistency. Please note that the attenuation factors in Sect. 3.2 refer to the attenuation of temperature variance measured with DTS, whereas text on P14L3 refer to the below-canopy vertical turbulent transfer of temperature. In general different size of eddies contribute differently to scalar variance and vertical turbulent flux (compare scalar power spectra and cospectra e.g. in (Kaimal and Finnigan, 1994)) and this discrepancy between scalar power spectra and cospectra is likely accentuated in below-canopy flows (Vickers and Thomas, 2013, 2014). In order to clarify the text, we will add a note on P12L11 that this part refers to variability of scalars only (and hence variance) and emphasize on P14L3 the influence of horizontal sensor separation rather than the characteristics of below-canopy turbulent transfer.

CHANGES TO THE MANUSCRIPT: changed "variability" on p12l11 to "scalar variability (and hence variance)" and changed the order of 1) and 2) on p14 and modified the text related to former 1) as "forest floor is close which suppresses the size of eddies dominating heat transfer which is also influenced by the canopy elements breaking the coherency of large eddies (spectral short circuit; Finnigan 2000; Launiainen et al. 2007) and DTS cannot capture the small scale turbulence". Furthermore, we continue the sentence ending at p14l10 with "and large horisontal sensor separation".

**Fig6 and other places: you are using the temperature from the sonic anemometers. I guess you converted sonic temperature in "real" temperature before use, state this somewhere. General question to the use of temperatures derived via sonic anemometer: they are not very accurate when it comes to absolute numbers, how did you deal with this? For prospective studies it might be more feasible to use profiles of e.g. thermocouples for comparison with the DTS. I also wonder about the Gill sonic anemometer: did you have any issues with this sonic regarding noise? I heard from several cases that the Gill sonic type you used has a problem with noise, also have personal experience with that issue. How did you deal with noise in the sonic data, if you observed it?**

RESPONSE: In fact sonic temperature was not converted to air temperature since the $H_2O$ concentration fluctuations needed for this conversion were not available at most heights (sonic temperature is almost equal to virtual temperature, which requires vapor and barometric pressures for computation of dry bulb 'normal' temperature). We failed to mention this deficiency in our measurements but will add it to Sect. 2.2 and also to Sect. 3.3 where DTS statistics are compared

against 3D sonics. This causes a minor bias in temperature variance (mainly depending on H2O variance) and w'T' covariance (depending on w'H2O' covariance) calculated from the sonic data (Foken et al., 2012). The referee is right the absolute numbers from sonic anemometers are not accurate, but fluctuations around the means are precise. However, please note that we did not use absolute temperatures from the sonics in DTS validation, but the mean T profiles from DTS were compared against slow-response thermometers (see Sect. 2.1) which should be accurate. We did not estimate noise in sonic data with the method presented in Sect. 2.2.1. However, based on power spectra (Fig. 3 in the manuscript) and visual evaluation of sonic raw data (see examples in Figs. 8 & 10 in the manuscript) the noise was only a minor component of the sonic anemometer measurements.

CHANGES TO THE MANUSCRIPT: Added the following sentence to p5l31: "Note that as $H_2O$ fluctuations were not measured at most heights, it was not possible to convert the sonic temperature to actual temperature (Schotanus, 1983; Foken et al., 2012). This caused a slight $H_2O$ variance and $H_2O$ flux dependent bias in temperature variance and $\overline{w'T'}$ covariance, respectively.". The following sentence was added to p12l15: "Note that statistics derived from 3D sonic anemometers were calculated from sonic temperature and hence slightly biased by $H_2O$ fluctuations (see Sect. 2.2)."

**Fig7: reformulate the figure caption in such way that it gets very clear that you applied the Cava parameterization only for unstable conditions: i.e., "For unstable conditions,.was also calculated based on Monin-Obukhov......".**

RESPONSE: Thanks, will be modified.

CHANGES TO THE MANUSCRIPT: replaced "$\sigma_T$ /|$T_*$| was also calculated based on Monin-Obukhov similarity theory (dashed lines) for unstable stratification using $\sigma_T$ /|$T_*$| = 2.3 (1+9.5|$\zeta$|)$^{-1/3}$ (Cava et al., 2008)" with "For unstable conditions $\sigma_T$ /|$T_*$| was also calculated based on Monin-Obukhov similarity theory (dashed lines) using $\sigma_T$ /|$T_*$| = 2.3 (1+9.5| $\zeta$ |)$^{-1/3}$$ (Cava et al., 2008)"

**Fig8: the arrows are very hard to identify.**

RESPONSE: Thanks, we will try to improve the figure. However, we struggled with the arrows already before submission and the figures in the manuscript were the best that we were able to come up with. Hence, big improvements in the figure are not likely, but we will try our best. The arrows help to illustrate the connection between the observed large temperature structures and wind fluctuations at different heights and hence we would like to keep them in the figure despite being somewhat unclear.

**P17: regarding vertical coupling: I would love to see here a comparison between the conclusions regarding coupling/decoupling which you derive with DTS and other approaches, e.g. the correlation of standard deviation of vertical wind (cf. Thomas et al., 2013). This does not have to be a complete analysis, this is beyond the scope of your paper, but a figure comparing coupling/decoupling conclusions derived via DTS and the mentioned correlation of sigma(w) would be very valuable, also for prospective work in this area. You have the required instrumentation there, a couple of 3-D sonics in profile.**

RESPONSE: Thank you for this comment, this is an interesting idea which we are exploring in a follow-up study. This manuscript deals with validating DTS measurements against conventional measurements at the forest-air interface and the suggested analyses on vertical coupling/decoupling are beyond the scope of this study.

**Conclusions: you touch very briefly the topic advection there; can you write to this a bit more? A vertical profile would not be sufficient for approaching advection issues, but a 3-D array could. How would such a setup in your opinion have to look like to approach advection sufficiently?**

RESPONSE: Yes, for properly addressing the question related to advection one likely needs to resolve the whole 3-D flow field within the forest. This is a formidable task for experimentalists and hence the problem related to advection has eluded a complete answer. Due to the spatially continuous measurements along the cables, horisontally and vertically distributed DTS cables would be able to resolve the mean temperature field and turbulent (up to 1 Hz) temperature fluctuations within the canopy. If these would be accompanied with actively heated fibre optics measuring wind (Sayde et al., 2015), the spatial variation of horizontal and vertical heat fluxes could be presumably approximated. This setup would of course miss the spatial variability of gases such as $CO_2$, but should still be one step closer to closing the ecosystem mass balance since it would shed light on the spatial variability of turbulent mixing within the canopy. Work is on its way regarding this aspect. We will add a note about this to the conclusions.

CHANGES TO THE MANUSCRIPT: Added the following sentence on p22l19: "This issue could be explored with DTS measurements utilising horizontally and vertically distributed fibre optic cables"

**Anonymous Referee #3**

**Overall, the manuscript presents the usefulness of a new and increasingly important tool with respect to spatial measurements of turbulence and atmosphere-ecosphere interactions. The manuscript is generally well put together but are a few places where some clarification is needed (see minor/technical comments below) to improve the message being presented. It is an interesting concept that adds to the toolbox of measurement techniques beyond the traditional flux tower setup to investigate the underlying turbulence that drives scalar fluxes in different landscapes and glad to see if developed further. Across the manuscript, the spatial extent/direction of the measurement needs to be clear. My only broader comment is in relation to the connection between the flow and temperature variability; the connection between these two values needs to be better supported; there are places where it appears the two variables are used interchangeably and it can lead to a little confusion for the reader (Sections 3.4). This is primarily an organizational issue, not a science issue.**

RESPONSE: We thank the referee for acknowledging the value of our study.

**Pg2ln6: "practical point-of-view" meaning a sufficient amount of spatially distributed tower measurements?**

RESPONSE: Yes, exactly. Constructing an array of measurement devices that capture all the relevant spatial details of the flow is a big and expensive undertaking. We argue that DTS measurements could help in this respect.

**Pg2ln6: "Hence..." Not sure how many spatial stats are being derived directly from time series observations at a single point; this needs to be clarified; its point is a little convoluted. I will agree that eddy structure, particularly in the vertical, is assumed from time series using Taylor's hypothesis but I don't necessarily agree that spatial statistics are being derived through Taylor's hypothesis.**

RESPONSE: Thanks for this comment, this needs clarification, since our wording was not accurate. What we mean with this part is that spatial details (eddy structures etc) of the flow are often evaluated from time series based on Taylor's frozen turbulence hypothesis, like the referee suggests. At the same time, it is often assumed that temporal and spatial statistics converge to ensemble statistics, i.e. the ergodic hypothesis is valid (Higgins et al., 2013) without having the experimental/ observational means to verify this assumption. We will modify this part of the manuscript based on this, the previous and the next comment.

CHANGES TO THE MANUSCRIPT: The first paragraph of the manuscript (p2l2…p2l10) was rewritten as "The majority of the interaction between the atmosphere and Earth's surface takes place in a shallow air layer termed the atmospheric boundary layer (ABL). Insights on the atmospheric mixing processes in this layer are required in order to gain a better understanding on ecosystem-atmosphere feedbacks, air quality and weather forecasting related issues. Studies near the surface typically rely on time series analysis, since spatial details of the mixing close to the ground are difficult to measure with conventional instrumentation. However, similarity theories underlying the analysis of observations and models are posed in length scales calling for spatially explicit sampling. In addition, turbulence statistics or spatial details of different flow structures are typically derived from a time series of observations by assuming ergodic hypothesis or Taylor's frozen turbulence hypothesis (Taylor, 1938), respectively, yet it is recognised that these hypotheses are not universally valid (Mahrt et al., 2009; Thomas, 2011; Higgins et al., 2012; Higgins et al., 2013; Cheng et al., 2017)."

**Pg2Ln8: "Another motivation…" feels tacked on; could be better framed to fit within the overall context of the need for more spatially explicit measurements.**

RESPONSE: Thanks, we will try to improve this part, see above.

**Pg3Ln23: "Furthermore, we evaluate…" remove "also" from this sentence.**

CHANGES TO THE MANUSCRIPT: done

**Pg3Ln25: "Deviations from Gaussian distribution are…"; assuming distribution of temperature? Clarify.**

RESPONSE: This refers to turbulence variables in general (e.g. wind components, temperature, gases), since their distribution typically deviates from Gaussian in the presence of large scale coherent patterns. See e.g. profiles for third-order statistics in canopy flows in LES study by (Patton et al., 2015).

**Pg5Ln19-23: "When comparing the results…" Not sure these two sentences are needed; there isn't that strong of a comparison back to the Thomas et al., 2012 paper within the manuscript. Though this work is based off the Thomas et al., work, unless a direct comparison within this paper is going to be made, the differences in the instrument variant is does not need to be explicitly stated.**

CHANGES TO THE MANUSCRIPT: This part was removed from the manuscript in response to this comment.

**Pg6Ln16-19: "After determining the differential attenuation…" I suggest splitting this into two sentences at the semicolon instead of keeping it as one longer, complex sentence since it is detailing two distinct processing steps.**

CHANGES TO THE MANUSCRIPT: Text was modified as the referee suggests.

**Pg6Ln23: "After quality filtering..." Out of how many potential 30-minute periods, either as a percent or total measurement periods?**

RESPONSE: 89 % of the total 30-min periods were available for analysis after quality filtering. We noticed that there was a typo in the amount of data available for analysis. It was 1513, not 1353 as previously mentioned in the manuscript.

CHANGES TO THE MANUSCRIPT: Modified the sentence as "After quality filtering, 1513 (89 % of the whole period) 30-min periods of DTS data were available for further analysis."

**Figure 3: Put the height of the sonic anemometer within the figure caption; easy to miss being the title of the graph.**

RESPONSE: Thanks, will be added.

CHANGES TO THE MANUSCRIPT: Changed the first sentence of Fig. 3 caption as "Ensemble-averaged, frequency weighted and normalised temperature power spectrum estimated from the DTS data and colocated 3D sonic anemometer (reference) at 25 m height above ground ($1.5h_c$, where $h_c$ is canopy height)."

**Pg12Ln16: "The temperature variance was dominated..." This feels out of place; there isn't any direct support for this comment near this sentence and think it should be later in the paragraph/section after the discussion of Figure 5/Table 2 or in the previous section with the discussion of the power spectra comparisons. Also strikes me as being a statement that applies to above the canopy and not as much within the canopy due to the general size of eddies closer to the ground and surrounded by obstacles (trees).**

RESPONSE: OK, we will remove this sentence and modify the next sentence so that it discusses only the power spectra and not the large coherent eddies.

CHANGES TO THE MANUSCRIPT: Removed the sentence and modified the next sentence as "Bulk of the time series variance was related to fluctuations close to the peak of the power spectra and DTS system was able to resolve the variability at these frequencies (Fig. 3)"

**Pg14Ln5: "size eddies dominating..." should be "size of eddies dominating..."**

CHANGES TO THE MANUSCRIPT: Thanks, modified as suggested.

**Pg15Ln7: Remove the "but" from "The mean potential temperature gradients from DTS bug exhibited...".**

CHANGES TO THE MANUSCRIPT: Thanks, removed.

**Pg16Ln4: "For comparison, the unbiased median temperature gradients...", Gradient between which heights; across canopy?**

RESPONSE: Between the heights mentioned in the previous sentence (27 m and 5.8 m).

CHANGES TO THE MANUSCRIPT: Modified the sentence as "For comparison, the unbiased median temperature gradients during these periods calculated from the reference measurements at heights mentioned above were 0.34 K and -0.29 K at night and daytime, respectively.".

**Pg16Ln15: Remove "already" in "hence a smaller..." sentence.**

CHANGES TO THE MANUSCRIPT: Removed

**Pg16Ln33: The word "almost is misspelled.**

CHANGES TO THE MANUSCRIPT: Thanks, fixed.

**Pg17Ln6-8: "The measured skewness profiles…" There is little to no context for this sentence as it requires the reader to look up these papers or be very familiar with them to understand the similarities in the skewness profiles. Clarify or add context to the sentence or remove.**

CHANGES TO THE MANUSCRIPT: OK, removed this sentence.

**Pg18Ln8-9: It might be better to mention of the caveat using Taylor's frozen turbulence hypothesis on this analysis at the beginning of the section instead of in the middle of the paragraph, or break the paragraph in half to be clear which pieces of the analyses are relying on Taylor's hypothesis.**

RESPONSE: Thanks, good suggestion.

CHANGES TO THE MANUSCRIPT: Replaced "eddy size" with "turbulence time scale" on p18l2. Split the paragraph into two starting from p18l4 and added in the beginning of the new paragraph the following sentence: "By relying on Taylor's frozen turbulence hypothesis, two dimensional spatial details of the coherent motions can be delineated from the vertical DTS measurements.". Removed the sentence about Taylor's hypothesis on p18l6…p18l8.

**Pg18Ln32: "Flow patterns…" This sentence is not saying anything new that the next sentence is not already saying and is not a very useful sentence. Maybe combine the two.**

CHANGES TO THE MANUSCRIPT: OK, removed this sentence.

REFERENCES

Foken, T., Leuning, R., Oncley, S. R., Mauder, M. and Aubinet, M.: Corrections and Data Quality Control BT - Eddy Covariance: A Practical Guide to Measurement and Data Analysis, edited by M. Aubinet, T. Vesala, and D. Papale, pp. 85–131, Springer Netherlands, Dordrecht., 2012.

Higgins, C. W., Katul, G. G., Froidevaux, M., Simeonov, V. and Parlange, M. B.: Are atmospheric surface layer flows ergodic?, Geophys. Res. Lett., 40(12), 3342–3346, doi:Doi 10.1002/Grl.50642, 2013.

Kaimal, J. C. and Finnigan, J. J.: Atmospheric boundary layer flows : their structure and measurement, Oxford University Press, New York. [online] Available from: http://helka.linneanet.fi/cgi-bin/Pwebrecon.cgi?BBID=636956, 1994.

Langford, B., Acton, W., Ammann, C., Valach, A. and Nemitz, E.: Eddy-covariance data with low signal-to-noise ratio: time-lag determination, uncertainties and limit of detection, Atmos. Meas. Tech., 8(10), 4197–4213, doi:10.5194/amt-8-4197-2015, 2015.

Mauder, M., Cuntz, M., Druee, C., Graf, A., Rebmann, C., Schmid, H. P., Schmidt, M. and Steinbrecher, R.: A strategy for quality and uncertainty assessment of long-term eddy-covariance measurements, Agric. For. Meteorol., 169, 122–135, doi:10.1016/j.agrformet.2012.09.006, 2013.

Nakai, T., Hiyama, T., Petrov, R. E., Kotani, A., Ohta, T. and Maximov, T. C.: Application of an open-path eddy covariance methane flux measurement system to a larch forest in eastern Siberia, Agric.

For. Meteorol., 282–283, 107860, doi:https://doi.org/10.1016/j.agrformet.2019.107860, 2020.

Nemitz, E., Mammarella, I., Ibrom, A., Aurela, M., Burba, G. G., Dengel, S., Gielen, B., Grelle, A., Heinesch, B., Herbst, M., Hörtnagl, L., Klemedtsson, L., Lindroth, A., Lohila, A., McDermitt, D. K., Meier, P., Merbold, L., Nelson, D., Nicolini, G., Nilsson, M. B., Peltola, O., Rinne, J. and Zahniser, M.: Standardisation of eddy-covariance flux measurements of methane and nitrous oxide, Int. Agrophysics, 32(4), doi:10.1515/intag-2017-0042, 2018.

Patton, E. G., Sullivan, P. P., Shaw, R. H., Finnigan, J. J. and Weil, J. C.: Atmospheric Stability Influences on Coupled Boundary Layer and Canopy Turbulence, J. Atmos. Sci., 73(4), 1621–1647, doi:10.1175/JAS-D-15-0068.1, 2015.

Peltola, O., Hensen, A., Helfter, C., Belelli Marchesini, L., Bosveld, F. C., van den Bulk, W. C. M., Elbers, J. A., Haapanala, S., Holst, J., Laurila, T., Lindroth, A., Nemitz, E., Röckmann, T., Vermeulen, A. T. and Mammarella, I.: Evaluating the performance of commonly used gas analysers for methane eddy covariance flux measurements: the InGOS inter-comparison field experiment, Biogeosciences, 11(12), 3163–3186, doi:10.5194/bg-11-3163-2014, 2014.

Rannik, Ü., Peltola, O. and Mammarella, I.: Random uncertainties of flux measurements by the eddy covariance technique, Atmos. Meas. Tech., 9(10), 5163–5181, doi:10.5194/amt-9-5163-2016, 2016.

Sayde, C., Thomas, C. K., Wagner, J. and Selker, J.: High-resolution wind speed measurements using actively heated fiber optics, Geophys. Res. Lett., 42(22), 10,10-64,73, doi:10.1002/2015GL066729, 2015.

Vickers, D. and Thomas, C. K.: Some aspects of the turbulence kinetic energy and fluxes above and beneath a tall open pine forest canopy, Agric. For. Meteorol., 181, 143–151, doi:https://doi.org/10.1016/j.agrformet.2013.07.014, 2013.

Vickers, D. and Thomas, C. K.: Observations of the scale-dependent turbulence and evaluation of the flux–gradient relationship for sensible heat for a closed Douglas-fir canopy in very weak wind conditions, Atmos. Chem. Phys., 14(18), 9665–9676, doi:10.5194/acp-14-9665-2014, 2014.

---

## Referee Report (RR1)

The manuscript "Suitability of fiber-optic distributed temperature sensing to reveal mixing processes and higher-order moments at the forest-air interface" reads well in the current state and presents a substantial scientific contribution to the community, highlighting the applicability of distributed temperature sensing on a variety of application fields.

After considering a number of minor, rather technical, suggestions (see below), I think the manuscript is ready to be published.
* * *
General: check the whole text for the use of articles, they are missing regularly.

**Abstract**

line 12: put a comma before and after "however"

line 16: replace "discern" with "be discerned"

**page 2**

line 9: ...respectively. Yet, it is...

line 16: information on

line 26: why is Traeumer in capital letters?

line 27: put a comma after yet?

line 29: replace yet with but. Split sentence in two: "......(e.g. at measurement towers). Consequently, spatial details..."

line 30: Hence,...

**page 3**

line 16: what is meant with transition periods?

line 20: systems

line 30: split sentence. ".....to boundary layer top. Consequently, the DTS system bridges...."

**page 4**

line 10: remove "the" before north-south direction. Write North and South in capital letters.

question for my interest: why do you have three different sonic types along the tower?

line 27: name the range of the lidar

**page 5**

line 12: reformulate "This setup enabled reference measurements at both...."

line 13: ".., and provided"

line 20: shouldn`t the mean be constant for each time averaging interval to ensure a proper application of this composition...?

**page 6**

line 6: "This value" refers to $u*$ or the $T*$?

line 10: "as a function..."

line 22: one bracket missing after the numbers

line 30: "...be approximated by"

**page 7**

lines 1-3: "An estimate for the attenuation of the DTS-derived temperature variance due to imperfect high frequency response and lower sampling frequency can be derived by"

line 28: comma after "Similarly"

**page 8**

line 20: the experimental setup

line 21: comma after hence?

line 25: split sentence; "...outer cable (sheath, protection). These are...."

line 27: estimate

line 28: due to the fibre-optic cable

**page 9**

line 9: "showed" instead of "did show"

**page 11**

line 5: ...can be considered independent...

line 8: you use "dependence" at several other places in the manuscript, here you use "dependency". Be consistent.

line 9:...can be considered constant...

**page 12**

table 1 heading: ...derived via...

line 1: comma after hence

line 3: comma after hence

line 4: ...a fixed value...

line 7: replace "did cause" with "caused"

**page 13**

line 1: comma before "suggesting"?

Figure 5: write wT in the figure with overbar and apostrophes, as it is standard to describe the turbulent fluxes

**page 14**

line 2: the sensor separation between DTS and sonic was the same above and below canopy, right...? if so, it cannot be used as argument here.

lines 7 - 10: split this sentence in two.

line 18 - page 15 (line 2): split this sentence in two.

**page 15**

first paragraph: this is a quite big difference in the temperature profiles between DTS and reference. I wonder if the authors want to say a sentence about strategies to improve DTS in this regard. Without considering the radiation errors DTS seems useless with regards to T profiles.

**page 16**

line 17: site (it`s only one)

line 18: ...a previous study...

line 18 - line 21: split sentence and reformulate.

line 22 - 24: not sure if everyone will understand what you try to say with this sentence

line 27: in which regard are they in line with previous studies...?

line 30: fluxes are constant with height only in the surface layer, and this surface layer is not very big...; furthermore, here we are in and above a forest which modifies the atmospheric stratification schemes.

**page 17**

line 15: ...the vertical what? Feels like here is a noun missing.

**page 18**

line 18: replace "do demonstrate" with "demonstrate"

line 22: to which surface are you referring here? soil surface? canopy top?

**page 19**

line 1: comma after "in principle"

**page 20**

fig. 9: it might be useful to indicate in this fig. that blue is standing for the unstable stratification example and pink for the stable stratification example. Maybe switch colours as blue is often used intuitively for stable stratification.

**page 21**

fig. 11: it would help the reader to name directly in this fig. that you are referring here to the stable stratification nighttime example. Also, it would help to indicate how gradients were calculated (upper height - lower height, I guess).

line 5: in line with what from Thomas et al. (2012)? And what in Thomas et al. (2012) is complemented? Readers will not have the content of Thomas et al. (2012) in mind. Reformulate: "...these results are in line with their findings and complement them."

line 14: in the current study...

**page 22**

line 9: ...help to separate...

line 10: ...since the measurements allow the tracking of the temperature...

line 16: place a comma after "ultimately"

Generally to the conclusions and outlook:

can the authors briefly acknowledge the shortcomings of DTS which got obvious in this study and give information on how these can be overcome? I have e.g. in mind the topic radiation error which appears to be quite substantial. And which makes the use of derived absolute temperatures questionable. Give a brief overview please on the current limitations of DTS and how they will be tackled in the future.

---

## Author Response (AR2)

Manuscript: Title: Suitability of fiber-optic distributed temperature sensing to reveal mixing processes and higher-order moments at the forest-air interface

Author(s): Olli Peltola et al.

MS No.: amt-2020-260

MS type: Research article

Review round 2

We thank both the editor and the referee for the comments on the revised version of the manuscript. Please find below point-by-point responses to the presented critique. Editor and referee comments are in **bold**, responses with red and changes to the manuscript with blue.

**EDITOR COMMENTS**

**1) In the previous review, referee#3 placed the following comment:**

**"My only broader comment is in relation to the connection between the flow and the temperature variability; the connection between these two values needs to be better supported; there are places where it appears that the two variables are used interchangeably and it can lead to a little confusion for the reader."**

**This comment has not been addressed in the auther's reponse. Please do so now.**

**(One exemple for the issue is on page 17, line 10)**

RESPONSE: We are sorry for missing this comment during the previous round and will answer to it now. While it is true that air movement cannot be directly derived from scalar- temporal and spatial variability, there is a large body of studies showing that particularly for forest canopy flows with well-developed coherent structures (CS) the signals between vectorial components and scalars are very systematic. This is why CS are considered part of the organized turbulence, and not part of the stochastic inertial subrange turbulence. Due to the mediating effect of the 3-D turbulent pressure field, structures observed in time series of vectorial components are not as sharp as those of scalars, this is also well documented (see e.g. Fig. 2 in (Thomas and Foken, 2005) or Fig.2 in (Thomas and Foken, 2007)). Hence, particularly for larger and longer-lived organized motions a direct inference between vector and scalar components seems justified. We will add a short note about this in the manuscript, but also emphasise that the analysis relies on temperature variability only.

CHANGES TO THE MANUSCRIPT: Modified Sect. 3.5 header as "Examples of organised patterns observed with the DTS system across vertical coupling regimes". Added "temperature" between "spatial" and "patterns" on page17line10. Added the following sentence to page17line11: "During well-developed turbulence these patterns are interlinked with large scale organised turbulent air motions (see e.g. Fig. 3 in Gao et al., 1989)." Modified the sentence on page17lines11-12 as "Large coherent eddies dominated the flow (Fig. 8b) and their signatures on vertical temperature profiles were captured with the DTS measurements." Changed the sentence on page18lines1-2 to "By relying on Taylor's frozen turbulence hypothesis, two dimensional spatial details of the large temperature patterns can be delineated from the vertical DTS measurements." Added "related to" before "downward" on page18line2. Replaced "turbulent motions" with "turbulence" on page18line19.

Added the following sentence on page20line2: "Note however, that these two examples relied only on temperature observations, for similar observations of wind vectors, actively heated fibre optics would be needed (Sayde et al., 2015)."

**2) page 2, line 9: omit "respectively"; change "it is recognised" to "it has been recognised"**

CHANGES TO THE MANUSCRIPT: Modified as suggested.

**3) page 6, line 18: clarify to "...time series for both parameters."**

CHANGES TO THE MANUSCRIPT: Modified as suggested.

**4) page 12, line 18: change to "The bulk of ..."**

**page 12, line 19: change to "...and the DTS system..."**

CHANGES TO THE MANUSCRIPT: Modified as suggested.

**5) page 16, line 3: omit "For comparison". (It is misleading here).**

**page 16, line 6: change to "As shown above, DTS and 3D sonic temperature measurements showed ..."**

CHANGES TO THE MANUSCRIPT: Modified as suggested.

**6) page 18, line 1: change to "two-dimensional"**

CHANGES TO THE MANUSCRIPT: Modified as suggested.

**Anonymous Referee #2**

**General: check the whole text for the use of articles, they are missing regularly.**

RESPONSE: We will try to improve the text in this respect.

**Abstract, line 12: put a comma before and after "however"**

CHANGES TO THE MANUSCRIPT: Added

**Abstract line 16: replace "discern" with "be discerned"**

CHANGES TO THE MANUSCRIPT: Modified as the referee suggests.

**Page 2line 9: ...respectively. Yet, it is...**

CHANGES TO THE MANUSCRIPT: Modified as the referee suggests.

**Page 2line 16: information on**

CHANGES TO THE MANUSCRIPT: Modified as the referee suggests.

**Page 2line 26: why is Traeumer in capital letters?**

RESPONSE: Due to issues related to the software used to manage references.

CHANGES TO THE MANUSCRIPT: Fixed.

**Page 2line 27: put a comma after yet?**

CHANGES TO THE MANUSCRIPT: Modified as the referee suggests.

**Page 2line 29: replace yet with but. Split sentence in two: "......(e.g. at measurement towers). Consequently, spatial details...”**

CHANGES TO THE MANUSCRIPT: Modified as the referee suggests.

**Page 2line 30: Hence,...**

CHANGES TO THE MANUSCRIPT: Modified as the referee suggests.

**Page 3, line 16: what is meant with transition periods?**

RESPONSE: Transition periods mean in these two studies sunrise in the morning (Higgins et al., 2018) and beginning and end of total solar eclipse (Higgins et al., 2019).

**Page 3, line 20: systems**

CHANGES TO THE MANUSCRIPT: Modified as the referee suggests.

**Page 3, line 30: split sentence. ".....to boundary layer top. Consequently, the DTS system bridges...."**

CHANGES TO THE MANUSCRIPT: Modified as the referee suggests.

**Page4line 10: remove "the" before north-south direction. Write North and South in capital letters. question for my interest: why do you have three different sonic types along the tower?**

RESPONSE: The different sonic types along the tower stem simply from the fact that the site has been running since 1996 and different sonic types have been purchased along the years. Furthermore, ICOS measurement protocols demand the usage of Gill HS-50 for EC measurements, whereas based on our own experiences METEK USA-1 sonic anemometers are maybe more suitable for northern conditions.

 CHANGES TO THE MANUSCRIPT: Modified the text as the referee suggests

**Page4line 27: name the range of the lidar**

CHANGES TO THE MANUSCRIPT: Added the following sentence: "The Doppler lidar provides data from 75 m to 9585 m in range and the clearest signal typically originates from the boundary layer where the aerosol loading is the highest."

**Page5line 12: reformulate "This setup enabled reference measurements at both...."**

CHANGES TO THE MANUSCRIPT: Modified the text as the referee suggests

**Page5line 13: ".., and provided"**

CHANGES TO THE MANUSCRIPT: Modified the text as the referee suggests

**Page5line 20: shouldn`t the mean be constant for each time averaging interval to ensure a proper application of this composition...?**

RESPONSE: In theory yes. For instance, air temperature (T) has a distinct diel course and hence T time series contain a low-frequency component which is not related to turbulence. This contamination of T' signal by non-turbulent motions can be counterbalanced by limiting the averaging period length or alternatively by using e.g. running mean filters (McMillen, 1988) to separate the turbulent signal from measurements. However, these approaches cause an underestimation of the turbulent variability at low-frequencies (Rannik and Vesala, 1999). Commonly accepted compromise is to use 30-min averaging period and block-averaging (Sabbatini et al., 2018).

**Page6line 6: "This value" refers to u* or the T*?**

CHANGES TO THE MANUSCRIPT: Replaced "This value" with "T*"

**Page6line 10: "as a function..."**

CHANGES TO THE MANUSCRIPT: Modified the text as the referee suggests

**Page6line 22: one bracket missing after the numbers**

CHANGES TO THE MANUSCRIPT: Fixed.

**Page6line 30: "...be approximated by"**

CHANGES TO THE MANUSCRIPT: Modified the text as the referee suggests

**Page7lines 1-3: "An estimate for the attenuation of the DTS-derived temperature variance due to imperfect high frequency response and lower sampling frequency can be derived by"**

CHANGES TO THE MANUSCRIPT: Modified the text as the referee suggests

**Page7line 28: comma after "Similarly"**

CHANGES TO THE MANUSCRIPT: Modified the text as the referee suggests

**Page8line 20: the experimental setup**

CHANGES TO THE MANUSCRIPT: Modified the text as the referee suggests

**Page8line 21: comma after hence?**

CHANGES TO THE MANUSCRIPT: Modified the text as the referee suggests

**Page8line 25: split sentence; "...outer cable (sheath, protection). These are...."**

CHANGES TO THE MANUSCRIPT: Modified the text as the referee suggests

**Page8line 27: estimate**

CHANGES TO THE MANUSCRIPT: Modified the text as the referee suggests

**Page8line 28: due to the fibre-optic cable**

CHANGES TO THE MANUSCRIPT: Modified the text as the referee suggests

**Page9line 9: "showed" instead of "did show"**

CHANGES TO THE MANUSCRIPT: Modified the text as the referee suggests

**Page11line 5: ...can be considered independent...**

CHANGES TO THE MANUSCRIPT: Modified the text as the referee suggests

**Page11line 8: you use "dependence" at several other places in the manuscript, here you use "dependency". Be consistent.**

CHANGES TO THE MANUSCRIPT: Replaced "dependency" with "dependence"

**Page11line 9:...can be considered constant...**

CHANGES TO THE MANUSCRIPT: Modified the text as the referee suggests

**Page12table 1 heading: ...derived via...**

CHANGES TO THE MANUSCRIPT: Modified the text as the referee suggests

**Page12line 1: comma after hence**

CHANGES TO THE MANUSCRIPT: Modified the text as the referee suggests

**Page12line 3: comma after hence**

CHANGES TO THE MANUSCRIPT: Modified the text as the referee suggests

**Page12line 4: ...a fixed value...**

CHANGES TO THE MANUSCRIPT: Modified the text as the referee suggests

**Page12line 7: replace "did cause" with "caused"**

CHANGES TO THE MANUSCRIPT: Modified the text as the referee suggests

**Page13line 1: comma before "suggesting"?**

CHANGES TO THE MANUSCRIPT: Modified the text as the referee suggests

**Figure 5: write wT in the figure with overbar and apostrophes, as it is standard to describe the turbulent fluxes**

CHANGES TO THE MANUSCRIPT: Modified the text as the referee suggests

**Page14line 2: the sensor separation between DTS and sonic was the same above and below canopy, right...? if so, it cannot be used as argument here.**

RESPONSE: We disagree with the referee and argue that the sensor separation can be use as an argument to explain the disagreement between below-canopy heat fluxes calculated using T either from DTS or 3D sonic anemometer measurements. Turbulence in the subcanopy is much more short-lived than above-canopy turbulence, and turbulence spectra can be multi-modal because of motions generated by waving branches, short-circuiting, von Karman vortices shed by tree trunks etc, see e.g. Fig.2 ( lower row at 4m in comparison with 16m and above-canopy level) in (Vickers and Thomas, 2014). Because of this short-lived nature at around 2 Hz and the weak winds, from Taylor's hypotheses one can infer that spatial scales are also small (assume time scale at peak of 2 Hz: t=0.5s; u=1m/s; l=u*t=0.5m) and hence sensor separation around these scales may have a significant impact on fluxes. Therefore, the flux (but not variance) related to these small eddies is damped due to large horizontal sensor separation.

**Page14lines 7 - 10: split this sentence in two.**

CHANGES TO THE MANUSCRIPT: Splitted the sentence in two: "This is supported by the finding that the in-canopy temperature variance was captured accurately, while the heat flux was not. The temperature fluctuations related to the large eddies sweeping into the in-canopy were captured accurately yet their signal was decorrelated with respect to the vertical wind speed (i.e. vertical turbulent flux) by the canopy elements and large horisontal sensor separation"

**Page14line 18 - page 15 (line 2): split this sentence in two.**

CHANGES TO THE MANUSCRIPT: Splitted the sentence in two: "Imposing a DTS SNR threshold of 0.5 still left 54 % of all DTS data measured during the campaign and 63 % of DTS data measured below 50 m height available for generating third-order statistics. Hence, we conclude that this DTS system can still be used after SNR filtering to monitor higher-order statistics and the non-Gaussian character of the flow despite the higher noise floor of this instrument relative to the older 2 km variant used in Thomas et al. (2012)."

**Page15first paragraph: this is a quite big difference in the temperature profiles between DTS and reference. I wonder if the authors want to say a sentence about strategies to improve DTS in this regard. Without considering the radiation errors DTS seems useless with regards to T profiles.**

RESPONSE: We partly agree with the referee, radiation errors significantly decrease the usability of mean temperature (T) measurements with DTS. However, please note that in this section of the manuscript we are analyzing T gradients, not mean values. The influence of radiation errors on gradients disappear if the errors are the same across the cable. However, this is not true when comparing above- and below-canopy data due to canopy shading and hence T gradients derived from DTS data were biased. However, we argue that the found biases in T gradients were not extremely large (0.07 K and 0.04 K at night and day, respectively) and hence do not fully invalidate the usage of these gradients in research, as the referee suggests. Given the very uniform longwave radiative environment in the subcanopy and the small shortwave mostly diffuse radiative fluxes (see e.g. Fig.2 in (Thomas, 2011)), the expected radiation errors in the below-canopy air space are on the order of a few tenth of K despite the weak flows. Sun flecks caused be direct-beam radiation do cause spatial variability, but magnitudes – again – are on the order of a few tenth of K for this specific fibre-optic (FO) cable. See more on the radiation errors for this specific FO cable type in (Sigmund et al., 2017). One should note that the radiation shielded Pt100 thermistors often used for air temperature measurements are not free from biases either (Huwald et al., 2009).

CHANGES TO THE MANUSCRIPT: Added the following sentence to page16line5: "Radiation shields around the cable could have presumably decreased these biases in gradients (Schilperoort et al., 2018), however the usage of screens would have invalidated the estimation of turbulent fluctuations from the DTS data since they disturb the turbulent air flow."

**Page16line 17: site (it`s only one)**

CHANGES TO THE MANUSCRIPT: Fixed.

**Page16line 18: ...a previous study...**

CHANGES TO THE MANUSCRIPT: Fixed.

**Page16line 18 - line 21: split sentence and reformulate.**

CHANGES TO THE MANUSCRIPT: Modified as "This is in line with a previous study at this site (Rannik, 1998) and others, where estimates for the roughness sublayer height typically range between $2h_c$ to $5h_c$ ($3h_c$ being the most common estimate) (Garratt, 1980; Coppin et al., 1986; Mölder et al., 1999; Poggi et al., 2004; Thomas et al., 2006)."

**Page16line 22 - 24: not sure if everyone will understand what you try to say with this sentence**

CHANGES TO THE MANUSCRIPT: We reformulated the sentence as "In near-neutral situations the scaled temperature variability ($\sigma_T/|T_*|$) exceeded the predictions made with M-O scaling, since the heat fluxes (and hence also $|T_*|$) decreased with $\zeta$ yet the temperature variability ($\sigma_T$) did not decrease at the same rate. In other words, heat transfer efficiency approached zero at the neutral limit (e.g. Rannik, 1998)."

**Page16line 27: in which regard are they in line with previous studies...?**

RESPONSE: Similarly as in here, Pahlow et al. (2001) showed that in strongly stable situations height is not a governing variable for $\sigma_T/|T_*|$ variability.

**Page16line 30: fluxes are constant with height only in the surface layer, and this surface layer is not very big...; furthermore, here we are in and above a forest which modifies the atmospheric stratification schemes.**

RESPONSE: Yes, vertical turbulent fluxes can be conjectured to be constant with height in the surface layer only, we refer to this with the "bottom part of the ABL" in the text. As the referee suggest, air flows above canopies differ significantly from boundary layer flows, but vertical turbulent fluxes can still be conjectured to be constant with height above canopies (Patton et al., 2015).

**Page17line 15: ...the vertical what? Feels like here is a noun missing.**

CHANGES TO THE MANUSCRIPT: Added "column" at the end of the sentence.

**Page18line 18: replace "do demonstrate" with "demonstrate"**

CHANGES TO THE MANUSCRIPT: Fixed.

**Page18line 22: to which surface are you referring here? soil surface? canopy top?**

RESPONSE: Surface here refers to the Earth's surface below the instrument. This includes mostly canopy but also those parts of the forest floor that are not covered by the canopy.

CHANGES TO THE MANUSCRIPT: Replaced "surface" with "forest"

**Page19line 1: comma after "in principle"**

CHANGES TO THE MANUSCRIPT: Fixed.

**Page20fig. 9: it might be useful to indicate in this fig. that blue is standing for the unstable stratification example and pink for the stable stratification example. Maybe switch colours as blue is often used intuitively for stable stratification.**

CHANGES TO THE MANUSCRIPT: Colors were switched based on the referee suggestion.

**Page21fig. 11: it would help the reader to name directly in this fig. that you are referring here to the stable stratification nighttime example. Also, it would help to indicate how gradients were calculated (upper height - lower height, I guess).**

RESPONSE: The method used to calculate the gradients is already described in the caption.

CHANGES TO THE MANUSCRIPT: Modified the first sentence of the Fig. 11 caption to "Evolution of potential temperature gradient (dθ/dz) (subplot a) for the nighttime period shown in Fig. 10 and concurrent wind speed timeseries from two heights (subplot b)."

**Page21line 5: in line with what from Thomas et al. (2012)? And what in Thomas et al. (2012) is complemented? Readers will not have the content of Thomas et al. (2012) in mind. Reformulate: "...these results are in line with their findings and complement them."**

RESPONSE: We opt not to give a detailed summary of the findings of Thomas et al. (2012) here. The agreement between the findings in this study and in Thomas et al. (2012) is already summarized in the first part of Sect. 4

CHANGES TO THE MANUSCRIPT: changed the sentence on page21lines5-6 to "Despite the fact that these results were obtained with a different DTS machine compared to Thomas et al. (2012) and that different cable suspension techniques were used, these results are in line with their findings on DTS capturing second-order moments of air temperature variability and complement them."

**Page21line 14: in the current study...**

CHANGES TO THE MANUSCRIPT: Replaced "In this study" with "In the current study"

**Page22line 9: ...help to separate...**

CHANGES TO THE MANUSCRIPT: Fixed

**line 10: ...since the measurements allow the tracking of the temperature...**

CHANGES TO THE MANUSCRIPT: Fixed

**line 16: place a comma after "ultimately"**

CHANGES TO THE MANUSCRIPT: Fixed

**Generally to the conclusions and outlook: can the authors briefly acknowledge the shortcomings of DTS which got obvious in this study and give information on how these can be overcome? I have e.g. in mind the topic radiation error which appears to be quite substantial. And which makes the use of derived absolute temperatures questionable. Give a brief overview please on the current limitations of DTS and how they will be tackled in the future.**

RESPONSE: The main shortcomings of the DTS measurement technique are 1) the high noise floor, 2) lower high frequency response compared to the 3D sonic anemometers and 3) the biases in absolute temperatures caused by radiation. The first two are already briefly mentioned in the first part of the Sect. 4 but radiation error is not mentioned. We will add a short sentence on this as well. There is not much to be done with the first two, since they are largely dictated by the DTS instrument specifications and the used fibre-optic cable. Radiation error on the other hand could be minimized by using radiation shields around the cable. However, like mentioned above, this would inhibit the estimation of turbulence from DTS measurements since the shields would strongly disturb the turbulent flow. We would like to emphasise that absolute temperatures (T) are rarely of great interest when studying mixing processes in the atmosphere (the topic of this manuscript). More important are T gradients and T variability as they are related to the controls on the turbulent mixing in the air. Furthermore, the influence of radiation errors on cross-canopy T gradients were not found

to be extremely large, especially when recognizing that traditional thermometers can be biased as well (Huwald et al., 2009).

CHANGES TO THE MANUSCRIPT: Added the following sentences to page21line9: "Cross-canopy temperature gradients were found to be biased due to radiation induced errors, especially during morning and evening. These biases could be reduced with radiation shields around the DTS cable, however such shields would severely disrupt the air flow and hence inhibit the estimation of turbulence from DTS measurements. Other approaches have been proposed as well (de Jong et al., 2015; Sigmund et al., 2017).". Modified also the sentence starting on page21line9 as: "Despite these shortcomings DTS measurements can provide the missing spatial details of atmospheric mixing close to the surface which cannot be acquired with conventional in-situ or remote sensing methods."

REFERENCES

Higgins, C. W., Wing, M. G., Kelley, J., Sayde, C., Burnett, J. and Holmes, H. A.: A high resolution measurement of the morning ABL transition using distributed temperature sensing and an unmanned aircraft system, Environ. Fluid Mech., 18(3), 683–693, doi:10.1007/s10652-017-9569-1, 2018.

Higgins, C. W., Drake, S. A., Kelley, J., Oldroyd, H. J., Jensen, D. D. and Wharton, S.: Ensemble-Averaging Resolves Rapid Atmospheric Response to the 2017 Total Solar Eclipse , Front. Earth Sci. , 7, 198 [online] Available from: https://www.frontiersin.org/article/10.3389/feart.2019.00198, 2019.

Horst, T. W. and Lenschow, D. H.: Attenuation of Scalar Fluxes Measured with Spatially-displaced Sensors, Boundary-Layer Meteorol., 130(2), 275–300, doi:10.1007/s10546-008-9348-0, 2009.

Huwald, H., Higgins, C. W., Boldi, M.-O., Bou-Zeid, E., Lehning, M. and Parlange, M. B.: Albedo effect on radiative errors in air temperature measurements, Water Resour. Res., 45(8), doi:https://doi.org/10.1029/2008WR007600, 2009.

McMillen, R. T.: An eddy correlation technique with extended applicability to non-simple terrain, Boundary-Layer Meteorol., 43(3), 231–245, doi:10.1007/BF00128405, 1988.

Patton, E. G., Sullivan, P. P., Shaw, R. H., Finnigan, J. J. and Weil, J. C.: Atmospheric Stability Influences on Coupled Boundary Layer and Canopy Turbulence, J. Atmos. Sci., 73(4), 1621–1647, doi:10.1175/JAS-D-15-0068.1, 2015.

Rannik, Ü. and Vesala, T.: Autoregressive filtering versus linear detrending in estimation of fluxes by the eddy covariance method, Boundary-Layer Meteorol., 91(2), 259–280, 1999.

Sabbatini, S., Mammarella, I., Arriga, N., Fratini, G., Graf, A., Hörtnagl, L., Ibrom, A., Longdoz, B., Mauder, M., Merbold, L., Metzger, S., Montagnani, L., Pitacco, A., Rebmann, C., Sedlák, P., Šigut, L., Vitale, D. and Papale, D.: Eddy covariance raw data processing for CO2 and energy fluxes calculation at ICOS ecosystem stations, Int. Agrophysics, 32(4), 495–515, doi:10.1515/intag-2017-0043, 2018.

Sigmund, A., Pfister, L., Sayde, C. and Thomas, C. K.: Quantitative analysis of the radiation error for aerial coiled-fiber-optic distributed temperature sensing deployments using reinforcing fabric as support structure, Atmos. Meas. Tech., 10(6), 2149–2162, doi:10.5194/amt-10-2149-2017, 2017.

Thomas, C. and Foken, T.: Detection of long-term coherent exchange over spruce forest using wavelet analysis, Theor. Appl. Climatol., 80(2), 91–104, doi:10.1007/s00704-004-0093-0, 2005.

Thomas, C. and Foken, T.: Flux contribution of coherent structures and its implications for the exchange of energy and matter in a tall spruce canopy, Boundary-Layer Meteorol., 123(2), 317–337, doi:10.1007/s10546-006-9144-7, 2007.

Thomas, C. K.: Variability of Sub-Canopy Flow, Temperature, and Horizontal Advection in Moderately Complex Terrain, Boundary-Layer Meteorol., 139(1), 61–81, doi:10.1007/s10546-010-9578-9, 2011.

Vickers, D. and Thomas, C. K.: Observations of the scale-dependent turbulence and evaluation of the flux–gradient relationship for sensible heat for a closed Douglas-fir canopy in very weak wind conditions, Atmos. Chem. Phys., 14(18), 9665–9676, doi:10.5194/acp-14-9665-2014, 2014.